# A pangenome reference of 36 Chinese populations

Yang Gao[1,2,3,4,27], Xiaofei Yang[5,6,7,27], Hao Chen[3,27], Xinjiang Tan[3,27], Zhaoqing Yang[8,27], Lian Deng[1,27], Baonan Wang[2], Shuang Kong[2], Songyang Li[2], Yuhang Cui[2], Chang Lei[1], Yimin Wang[3], Yuwen Pan[3], Sen Ma[3], Hao Sun[8], Xiaohan Zhao[2], Yingbing Shi[1], Ziyi Yang[1], Dongdong Wu[9], Shaoyuan Wu[10], Xingming Zhao[11], Binyin Shi[12], Li Jin[1,2], Zhibin Hu[13,14], Chinese Pangenome Consortium (CPC)*, Yan Lu[1✉], Jiayou Chu[8✉], Kai Ye[6,15,16✉] & Shuhua Xu[1,2,4,10,17,18✉]

Human genomics is witnessing an ongoing paradigm shift from a single reference sequence to a pangenome form, but populations of Asian ancestry are underrepresented. Here we present data from the first phase of the Chinese Pangenome Consortium, including a collection of 116 high-quality and haplotype-phased de novo assemblies based on 58 core samples representing 36 minority Chinese ethnic groups. With an average 30.65× high-fidelity long-read sequence coverage, an average contiguity N50 of more than 35.63 megabases and an average total size of 3.01 gigabases, the CPC core assemblies add 189 million base pairs of euchromatic polymorphic sequences and 1,367 protein-coding gene duplications to GRCh38. We identified 15.9 million small variants and 78,072 structural variants, of which 5.9 million small variants and 34,223 structural variants were not reported in a recently released pangenome reference[1]. The Chinese Pangenome Consortium data demonstrate a remarkable increase in the discovery of novel and missing sequences when individuals are included from underrepresented minority ethnic groups. The missing reference sequences were enriched with archaic-derived alleles and genes that confer essential functions related to keratinization, response to ultraviolet radiation, DNA repair, immunological responses and lifespan, implying great potential for shedding new light on human evolution and recovering missing heritability in complex disease mapping.

Over the past two decades, the reference human genome sequence has served as the foundation for genetic and biomedical research and applications; however, there is a broad consensus that no single reference sequence can represent the genomic diversity of global populations. On one hand, high-quality population-specific and haplotype-resolved genome references are necessary for genetic and medical analysis[2]. On the other hand, there is a clear need to shift from a single reference to a pangenome form that better represents genomic diversity, or allelic variation, within and across human populations[3]. With the

advancement of long-read sequencing technologies as well as computational methods, it is now feasible to enable pan-genomic construction to capture the missed variations from a large collection of diverse genomes[4]. The Human Pangenome Reference Consortium (HPRC) recently constructed a draft human pangenome reference based on 47 samples of worldwide populations, but with East Asian population samples underrepresented ($n$ = 4)[1]. In particular, only three Southern Han Chinese (CHS) samples were included in the HPRC reference, too few to represent the genomic diversity of ethnic groups

[1]State Key Laboratory of Genetic Engineering, Human Phenome Institute, Zhangjiang Fudan International Innovation Center, Center for Evolutionary Biology, School of Life Sciences, Fudan University, Shanghai, China. [2]Ministry of Education Key Laboratory of Contemporary Anthropology, Collaborative Innovation Center for Genetics and Development, Fudan University, Shanghai, China. [3]Key Laboratory of Computational Biology, Shanghai Institute of Nutrition and Health, University of Chinese Academy of Sciences, Chinese Academy of Sciences, Shanghai, China. [4]School of Life Science and Technology, ShanghaiTech University, Shanghai, China. [5]School of Computer Science and Technology, Faculty of Electronic and Information Engineering, Xi'an Jiaotong University, Xi'an, China. [6]MOE Key Lab for Intelligent Networks & Networks Security, Faculty of Electronic and Information Engineering, Xi'an Jiaotong University, Xi'an, China. [7]Genome Institute, The First Affiliated Hospital of Xi'an Jiaotong University, Xi'an, China. [8]Department of Medical Genetics, Institute of Medical Biology, Chinese Academy of Medical Sciences, Kunming, China. [9]State Key Laboratory of Genetic Resources and Evolution, Kunming Institute of Zoology, Chinese Academy of Sciences, Kunming, China. [10]Jiangsu Key Laboratory of Phylogenomics & Comparative Genomics, International Joint Center of Genomics of Jiangsu Province School of Life Sciences, Jiangsu Normal University, Xuzhou, China. [11]Institute of Science and Technology for Brain-Inspired Intelligence, Ministry of Education Key (MOE) Laboratory of Computational Neuroscience and Brain-Inspired Intelligence, MOE Frontiers Center for Brain Science Fudan University, Shanghai, China. [12]Department of Endocrinology, The First Affiliated Hospital of Xi'an Jiaotong University, Xi'an, China. [13]State Key Laboratory of Reproductive Medicine, Nanjing Medical University, Nanjing, China. [14]Jiangsu Key Lab of Cancer Biomarkers, Prevention and Treatment, Collaborative Innovation Center for Cancer Personalized Medicine, Center for Global Health, School of Public Health, Nanjing Medical University, Nanjing, China. [15]School of Automation Science and Engineering, Faculty of Electronic and Information Engineering, Xi'an Jiaotong University, Xi'an, China. [16]School of Life Science and Technology, Xi'an Jiaotong University, Xi'an, China. [17]Department of Liver Surgery and Transplantation Liver Cancer Institute, Zhongshan Hospital, Fudan University, Shanghai, China. [18]Center for Excellence in Animal Evolution and Genetics, Chinese Academy of Sciences, Kunming, China. [27]These authors contributed equally: Yang Gao, Xiaofei Yang, Hao Chen, Xinjiang Tan, Zhaoqing Yang, Lian Deng. *A list of authors and their affiliations appears at the end of the paper. ✉e-mail: lueyan@fudan.edu.cn; chujy@imbcams.com.cn; kaiye@xjtu.edu.cn; xushua@fudan.edu.cn

in a region such as China, which is populated by 1.44 billion people. We showed previously that the genetic diversity in Asia was not well covered by large-scale international collaborative projects such as the 1000 Genomes Project[5,6]. Although the need to improve the representation of diverse ancestral backgrounds in genomic research is well known[7,8], substantially fewer genomic studies have been conducted in populations of Asian ancestry compared with populations of European ancestry. China harbours a great genetic diversity, with 55 officially recognized minority ethnic groups in addition to the Han Chinese majority and a considerable number of unrecognized ethnic groups. Despite advances in sequencing technologies leading to the achievement of the telomere-to-telomere haploid assembly T2T-CHM13 (ref. 9), only a limited number of Chinese genomes have been de novo assembled to high-quality haplotype sequences using long-read DNA sequencing technologies[2,10–14]. The only two published studies on the Chinese pangenome were limited to the short-read sequencing data of Han Chinese samples[15,16]. There is an urgent need to establish a high-quality pangenome reference that better represents the great genomic diversity of Chinese populations. We anticipate such an effort to broaden the reference to represent genomic diversity, resolve allelic and locus heterogeneity, support unbiased and comprehensive detection of structural variation within and across populations, and improve genotyping accuracy in genomic regions enriched with complex sequence variations, such as human leukocyte antigen genes, and ultimately facilitate genomic analysis for both evolutionary and medical research.

The Chinese Pangenome Consortium (CPC) aims to de novo sequence at least 500 individuals to better detect and catalogue sequence variants of the Chinese populations using third-generation sequencing technologies. Here we present the draft Chinese pangenome reference based on the first sequencing effort (Phase I) of the CPC, reporting 116 high-quality de novo assemblies from 58 core samples representing 36 minority Chinese ethnic groups and 6 assemblies of the Han Chinese majority.

## Assembly of diverse Chinese genomes

The full set of CPC Phase I assemblies includes 68 samples sequenced to an average depth of 28.82× (14.29–60.67×) of PacBio high-fidelity (HiFi) long reads (Methods), 9 samples sequenced with linked reads, and 11 samples sequenced with Oxford Nanopore Technologies long reads and Hi-C Illumina short reads (Supplementary Table 1). After quality control (see below and Methods), we assembled 58 phased diploid assemblies with an average depth of 30.65× (21.43–60.67×) from CPC core samples representing 36 Chinese minority ethnic groups and 8 linguistic groups (Fig. 1a and Supplementary Fig. 1). We further incorporated five samples of high-coverage Oxford Nanopore Technologies long-read sequencing, five published assemblies of four Chinese populations[2,10–12] and two Han Chinese samples from Central China with Oxford Nanopore Technologies long-read sequencing (Fig. 1a, Supplementary Fig. 1 and Supplementary Table 1). The CPC core samples showed substantial genetic diversity and covered most of the Chinese ethnic minority groups (Fig. 1a and Supplementary Fig. 2).

We developed an analysis pipeline to carry out genome assembly, quality control and assembly polishing of the newly sequenced PacBio HiFi samples (Methods and Extended Data Fig. 1a). We conducted both the primary assembly and diploid assembly of 68 PacBio HiFi samples using Hifiasm software and assessed the quality of each assembly (Methods). We removed three samples from the primary assemblies and seven samples from the diploid assemblies with relatively low quality (Extended Data Fig. 1b,c). Eventually, 58 samples or 116 high-quality assemblies were retained for further analysis (Supplementary Table 2). We further carried out assembly correction of the 116 haploid assemblies of the 58 samples, and the comparison of assembly quality before and after polishing showed that our assembly correction improved the data quality (Supplementary Fig. 3).

We assessed the 116 assemblies with an average total genome length of 3.01 gigabases (Gb), ranging from 2.88 Gb to 3.12 Gb owing to the size difference between the sex chromosomes, and 93.1% of assemblies showed a larger genome length than the ungapped GRCh38 (2.94 Gb; Fig. 1b and Supplementary Table 2). The 116 assemblies contained an average of 690.5 contigs and ranged from 294 to 1,568 (Supplementary Table 2). The median contig length of genomic assembly (that is, contig N50) ranged from 11.66 Mb to 87.3 Mb across our 116 assemblies with an average of 35.63 Mb, and 8.62% of assemblies had contig N50 values greater than those of GRCh38 (57.88 Mb; Fig. 1b and Supplementary Table 2). We showed that more than 99.67% of circular consensus sequencing reads were remapped to the contigs in each assembly, and the average quality value of the 116 assemblies was 52.84, varying between 47.33 and 59.45 (Fig. 1c). On average, about 24.34 Mb (about 17.9–43.01 Mb) of contigs were not aligned to the T2T-CHM13 reference, and on average, about 69 Mb (about 53.32–89.05 Mb) of contigs were not aligned to the GRCh38 reference, indicating that the genome of the CPC samples was not fully covered by either of the two reference genomes (Fig. 1d). Our assembly covered 96.54% (92.55–98.03%) of the GRCh38 reference and 93.59% (89.66–95.77%) of the T2T-CHM13 reference (Fig. 1d), showing that some of the genomic regions of our assemblies were systematically unassembled or could not be reliably aligned, probably owing to highly repetitive regions. Furthermore, we annotated the unmapped regions of the contigs, and found that about 84% (about 39.7–75.1 Mb; 58.1 Mb on average) of the unmapped sequences were satellite repeats (Supplementary Table 3). We next applied Inspector to reveal that there was an average of 3,627 small-scale assembly errors (1,889–6,848) in each assembly, including 327 expansions (143–610), 454 collapses (154–898) and 2,846 substitutions (1,287–5,663), which decreased after assembly polishing (Fig. 1e and Supplementary Fig. 3). There was an average of 38 large-scale assembly errors (11–78) in each assembly (Supplementary Fig. 4). However, we found an interesting case in which the evaluation software is unable to determine the correct assembly when sequencing and assembly errors coexist (Supplementary Fig. 5), indicating the limitations of current evaluation tools when dealing with challenging situations. We finally assessed the duplication ratios relative to the GRCh38 and T2T-CHM13 references, respectively, which were approximately at the same level among the 116 assemblies of the 58 samples (Fig. 1f).

## Genomic features of the CPC assemblies

To investigate the genomic composition of our 58 CPC samples, we aligned the 116 assemblies to T2T-CHM13 and called the variants of each sample using Phased Assembly Variant caller[17] (Methods). We observed an average of 11.22% (9.35–13.05%) small variants and 24.16% (19.72–29.3%) structural variants (SVs) per CPC sample were in the highly repetitive regions (Extended Data Fig. 2), and more than half of the SVs were singletons except for those on the Y chromosome (Supplementary Fig. 6). We further identified an average of 2,802 (2,265–3,187) novel sequences for each CPC sample by collecting insertions larger than 1 kb compared to the T2T-CHM13 reference (Methods and Extended Data Fig. 3a). These novel sequences were located at 63,243 regions across the genome, and they were particularly enriched in the highly repetitive sequences such as the centromeric and telomeric regions (Extended Data Fig. 3b). We determined 115 genomic regions as insertion hotspots based on the insertion frequency, covering a total length of 204.8 Mb.

Considering that reference bias typically hinders the identification of novel SVs in both short- and long-read datasets, we further completed the analysis of complex structural variations (CSVs) using our CPC high-quality and phased haplotype sequences. On average, we identified 70 CSVs per sample with the newly developed method SVision[18] (Supplementary Fig. 7). The most common CSV type was INS+tDUP with an average number of 44 (Supplementary Fig. 8). We merged the CSVs into a set of 706 non-redundant CSVs (Extended Data

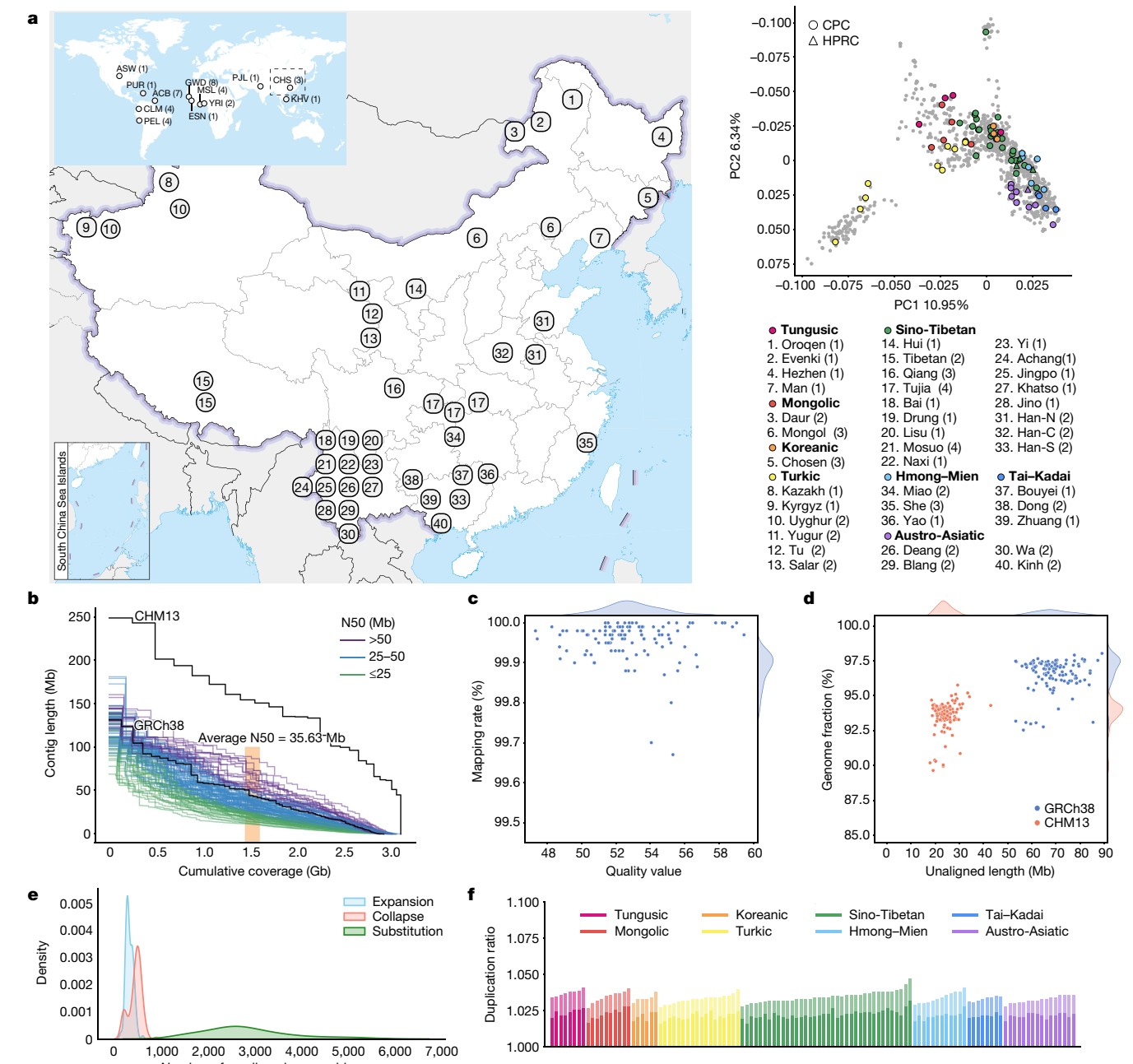

**Fig. 1 | CPC panel with diploid assemblies of 58 core samples. a**, Left: the geographical locations and ethnic, linguistic and genetic affiliations of the samples sequenced by CPC (see Supplementary Table 1 for details). The geographical distribution of HPRC samples is shown in the top left. Han-N, Han Chinese from North China; Han-S, Han Chinese from South China; Han-C, Han Chinese from Central China. Top right: the results of principal component (PC) analysis based on whole-genome data of the CPC samples (coloured dots) in the context of East Asian populations (grey dots). The four East Asian samples in HPRC are indicated using triangles in the principal component plot. **b**, NGx plot showing the assembly contiguity of the 116 CPC core assemblies. The contigs of T2T-CHM13 and GRCh38 (N-masked) are included for comparison. **c**, Assembly quality of the 116 CPC core assemblies. The x axis shows the quality value of each assembly, and the y axis shows the rate at which the circular consensus sequencing reads were remapped to the assembly contigs. **d**, Completeness of the 116 CPC core assemblies. The x axis represents the total length of the genome that could not be aligned to the GRCh38 and T2T-CHM13 references, and the y axis shows the proportion of each assembly aligned to GRCh38 and T2T-CHM13. **e**, Density plot showing the small-scale assembly error distribution of the 116 CPC core assemblies. **f**, Duplication ratio of the 116 CPC core assemblies. The dark and light colours of each bar represent the duplication rate related to T2T-CHM13 and GRCh38, respectively. The map of China in **a** was obtained from a standard map service (GS[2020]4618) approved by the Ministry of National Resources of the People's Republic of China (https://m.mnr.gov.cn).

Fig. 4a) and classified them into four classes[19]: shared (all samples), major (≥50% of samples), polymorphic (<50% of samples) and singleton (only one sample). We found that the peak of the CSV length distribution was at 10 kb (Extended Data Fig. 4b). We found one shared CSV (chr11:66,245,313–66,246,097 with an INS+INV type), 34 major CSVs, 277 polymorphic CSVs and 394 singleton CSVs from the CPC assembly set (Extended Data Fig. 4c).

We next annotated the copy number variations (CNVs) relative to the GRCh38 reference in each assembly (Methods). There were 1,367 protein-coding genes in the full set of assemblies that had a gain in

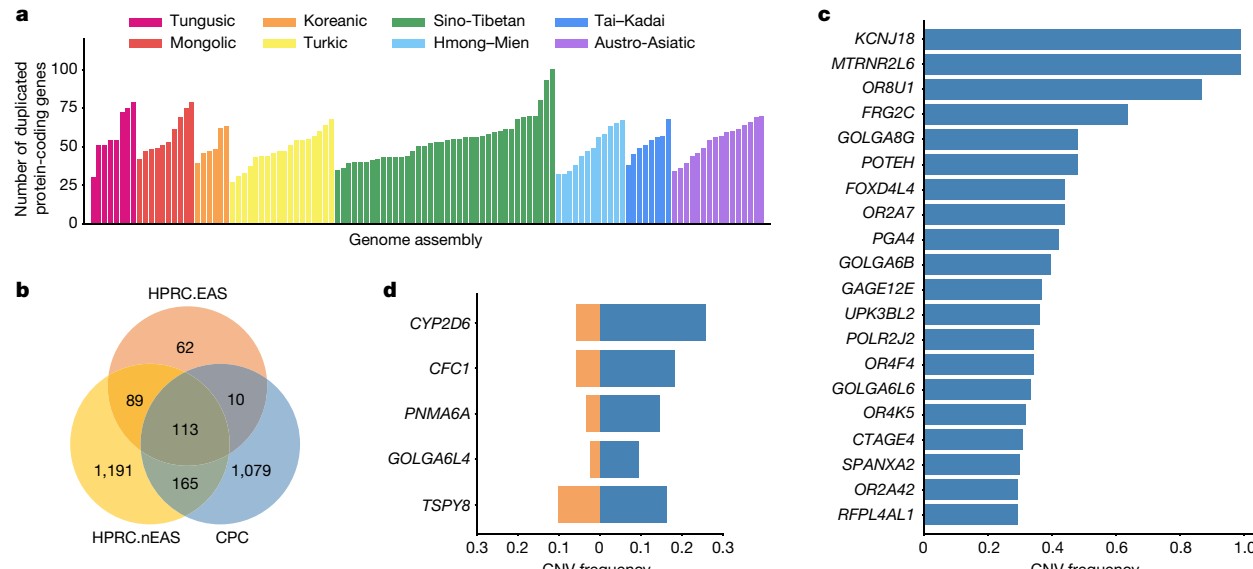

**Fig. 2 | CNVs identified from CPC assemblies. a**, Number of duplicated protein-coding genes per CPC genome assembly compared with the GRCh38 reference. **b**, Venn diagram showing the number of duplicated genes in CPC, HPRC.EAS and HPRC.nEAS assemblies. **c**, The top 20 most common CPC-specific CNV-related genes compared with the HPRC assemblies. **d**, The five overlapped CNV genes showing a higher frequency (≥5%) in CPC assemblies (blue) than in HPRC assemblies (orange). HPRC.EAS, East Asian in HPRC; HPRC.nEAS, non-East Asian in HPRC.

copy number in at least one genome. An average of 53 (27–100) genes with a gain in copy number were observed in each assembly (Fig. 2a), 13.39% of CNV genes showed a frequency of more than 5% in the whole CPC assembly set, and 57.86% of CNV genes were found in only a single haplotype. We found there were 1,079 duplicated genes in the CPC assembly set that were not observed in HPRC assemblies (Fig. 2b and Supplementary Table 4). These genes were enriched significantly in olfactory-related functions (for example, GO:0007608, sensory perception of smell, odds ratio (OR) = 6.46, Benjamini–Hochberg (BH)-adjusted $P = 2.22 \times 10^{-36}$; and hsa04740, olfactory transduction, OR = 7.14, BH-adjusted $P = 8.35 \times 10^{-38}$), and at a marginal significance level in skin-related disease (for example, DOID:37, skin disease, OR = 2.33, BH-adjusted $P = 0.086$; and DOID:16, integumentary system disease, OR = 2.23, BH-adjusted $P = 0.086$; Supplementary Table 5). We found that 562 (52.1%) of the novel duplicated genes in the CPC assembly set were trait-associated according to the genome-wide association study (GWAS) catalogue, and 207 (19.2%) of these genes were related to at least one human disease annotated by the Disease Ontology, suggesting the remarkable potential for the CPC assembly set to contribute to disease and phenotype association studies. In particular, *KCNJ18* and *MTRNR2L6* showed considerably high frequency in the CPC assemblies (115 of 116) but were absent in the HPRC assemblies (Fig. 2c). *KCNJ18* encodes a member of the inwardly rectifying potassium channel family (Kir2.6), which is expressed in skeletal muscle and is transcriptionally regulated by thyroid hormone[20]. Kir2.6 alterations were reported to be associated with thyrotoxic periodic paralysis, which is a well-known complication of thyrotoxicosis in Asian populations, but is rare in European populations[21–23]. The pseudogene *MTRNR2L6* is predicted to be involved in the regulation of apoptosis (GO:1900118). Notably, this gene was reported to show signatures of positive selection across geographical Han Chinese populations[24]. A number of the CPC-specific CNV genes showed signatures of natural selection indicated by the estimates of Tajima's $D$ (Supplementary Fig. 9 and Supplementary Table 4). Outstanding examples include *SPATA31D4* ($n_{haplotype} = 6$, Tajima's $D = -3.01$, false-discovery rate (FDR)-adjusted $P = 2.38 \times 10^{-5}$) and *PSAPL1* ($n_{haplotype} = 7$, Tajima's $D = 4.32$, FDR-adjusted $P = 0.006$). In particular, *SPATA31D4* belongs to the core duplicon families that are thought to have contributed significantly to hominoid evolution with

a strong signal of positive selection in hominoids. Previous studies suggested that the gene family has a function in response to ultraviolet radiation and DNA repair, and may also influence human lifespan[25].

There were 288 duplicated genes from the CPC assemblies that were also observed in the HPRC assemblies, of which 123 genes were shared with 4 HPRC East Asian samples (HPRC.EAS), and 278 genes were shared with the remaining 40 non-East Asian HPRC (HPRC.nEAS) samples (Fig. 2b). Among these overlapped genes, we found several genes with a higher frequency (≥5%) in the CPC assembly set than in the HPRC assemblies (Fig. 2d and Supplementary Table 6). For example, the CNV frequency at *CYP2D6* was higher in the CPC assembly set than those in the collective HPRC assemblies and in the separate sets of HPRC. EAS or HPRC.nEAS (Fig. 2d and Extended Data Fig. 5). *CYP2D6* is well known for its remarkable polymorphism, which has been systematically surveyed in a wide variety of human populations[26]. We also found *CFC1* with highly differentiated CNVs between the CPC and HPRC assemblies (frequency difference = 0.124). Several investigations reported associations between *CFC1* and cardiovascular disease[27], especially in Chinese populations[28]. In addition, *CTAG1A* and *ZNF658* were identified with rare CNVs (around 1%) in the HPRC assembly but presented with common CNVs (≥5% and <95%) in the CPC assembly. *CTAG1A* is expressed at high levels in the normal ovary and testis. *ZNF658* regulates the transcription of genes involved in zinc homoeostasis and affects ribosome biogenesis[29], and is involved in infectious processes (hsa05168, herpes simplex virus 1 infection).

## The variation graph of the CPC pangenome

In a typical pangenome reference, genomic data from a population can be organized into an edge-based sequence variation graph. We applied the Minigraph-Cactus pipeline to construct the variation graph for the CPC pangenome, in which the haplotypic assemblies can be represented as different paths composed of sequence nodes (Fig. 3a). Using Minigraph, 122 haplotypic assemblies from 61 samples were added to the graph starting with the reference assemblies T2T-CHM13 and GRCh38. To extend the pangenome to the single nucleotide polymorphism level, assemblies were remapped afterwards to the graph using the Cactus genome aligner. We used dna-brnn to

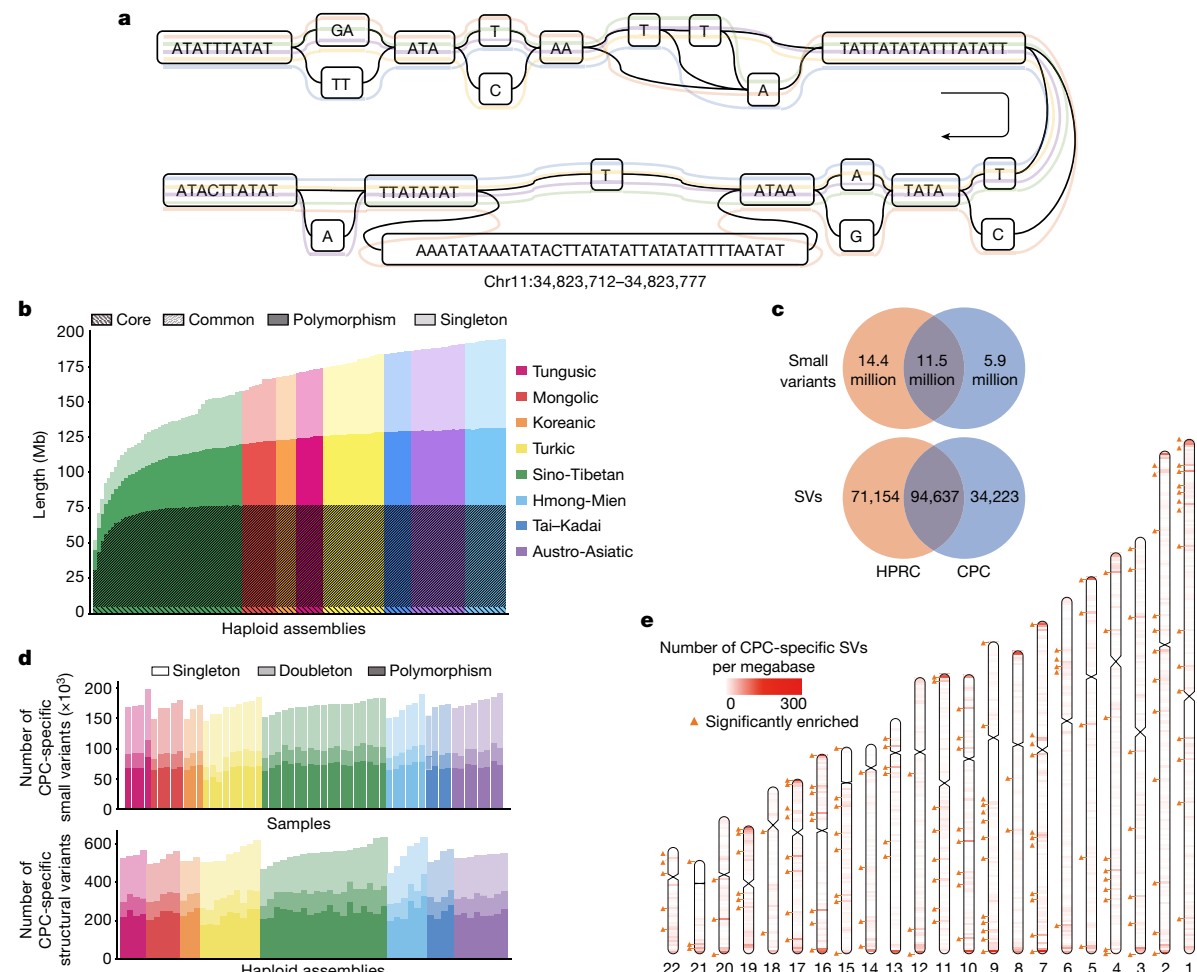

**Fig. 3 | CPC pangenome graph and CPC-specific variants compared to the HPRC assemblies. a**, A variation graph representing CHM13.chr11:34,823, 712–34,823,777 of the CPC pangenome. Coloured lines represent five haplotype assemblies. **b**, Pangenome cumulative growth curves for the CPC pangenome graph. Depth in bars measures how often a non-reference segment occurred in the assembled haplotypes. Non-reference sequences were classified into four categories according to the frequency of presence in the haplotypes: core (≥95%), common (≥5% and <95%), polymorphism (≥2 haplotypes but <5%) and singleton (only one haplotype). Different colours indicate a different ethnic group. **c**, The number of CPC-specific and common variants between CPC and HPRC assemblies from the joint pangenome graph. **d**, Number of identified CPC-specific small variants and SVs in different populations. **e**, The distribution of CPC-specific SVs from the joint pangenome graph on autosomes. The colour scale from white to red represents the density of SVs in 1-Mb regions. The orange triangle marks CPC-specific SVs that were significantly enriched in this 1-Mb window compared with HPRC-specific variants and common variants, on the basis of a one-tailed Fisher's exact test (FDR-correlated P value < 0.05).

identify and mask the centromeric satellites of the input assemblies, and then clipped all paths longer than 100 kb that were not aligned to the underlying graph to avoid the effects of complicated regions on alignment. Finally, we constructed the Minigraph-Cactus graph, which had a length of 3,284,609,818 bp (measured as the sum of all nodes). Additionally, we measured the length of the sequences that were added to the pangenome graph from each of the 61 diploid genomes. A total of 194.67 Mb of non-reference sequences were added to the graph, of which 62.90 Mb were singletons. In addition, we discovered that 4.96 Mb of non-reference sequences presented in ≥95% of all haplotypes, representing the core genome in our sample populations, and 72.24 Mb of non-reference sequences in ≥5% and <95% of the haplotypes, representing the common genome (Fig. 3b).

As the underlying 122 haplotypic assemblies were encoded as the paths in the graph, we characterized the variants as well as the haplotypes of assemblies by graph decomposition (Methods). We further examined whether the longest allele presented at each site, and we identified 15,916,404 small-variant sites (<50 bp) and 78,072 SV sites (≥50 bp) from the graph. Each sample contained 4,397,706 (s.d. = 35,776) small variants, and each haplotype contained 14,557

(s.d. = 224) SVs (Supplementary Fig. 10). The number of SVs obtained from each sample based on HiFi data was far more than that based on next-generation sequencing (NGS) data (Supplementary Table 7). The average length of SVs was 3.2 kb in the CPC graph genome, and the median length was 178 bp, which was longer than the short reads in the NGS data. In addition, we observed a peak at about 300 bp for Alu, and at about 6 kb for LINE-1 (Extended Data Fig. 6), indicating the reliability of our CPC pangenome.

The CPC core samples greatly covered the genetic diversity of China's multi-ethnic population and showed more diverse genetic ancestry than the Han Chinese samples (Extended Data Fig. 7). To evaluate the contribution of multi-ethnic populations in the CPC pangenome, we compared the cumulative length of the non-reference sequence by using Han Chinese samples alone as well as using multi-ethnic populations. When the first haplotype was added, the length of the non-reference sequence generated by the Han Chinese samples and that of the other populations were roughly the same (Extended Data Fig. 8). The average non-reference sequence length was 49.71 Mb for the Han Chinese and 49.16 Mb for other populations. However, we obtained remarkably different results with the addition of haplotypes. In particular, the

cumulative length grew much faster for the non-reference sequences detected in multi-ethnic populations than those detected in a single population, such as the Han Chinese (Extended Data Fig. 8).

## Short-read mapping with the CPC reference

To evaluate the performance of the Giraffe mapper to process graph genomes with different complexities, we applied gradient filtering on the CPC reference according to the path depth of nodes (Methods). When a more stringent filter was applied, we observed a decrease in the number of nodes and edges in the graph reference (Supplementary Fig. 11a), the graph complexity (Supplementary Fig. 11b) and the diversity (Supplementary Fig. 11c). Next, ten samples from the East Asian population of the 1000 Genomes Project were aligned to these CPC references with different complexities through vg Giraffe. The results showed that the mapping rate increased and reached a peak value with a simplified version of the graph (Supplementary Fig. 12a), probably owing to the limitation of the current version of the Giraffe mapper in managing the locally complex regions. We also observed that the proportion of reads with perfect matching continued to decline with the simplification of the graph (Supplementary Fig. 12b), reflecting the decline in the diversity of the graph (Supplementary Fig. 13). Therefore, there was a trade-off between mapping rate and mapping quality, and a compromise is needed to determine the size of the graph reference.

Compared with the HPRC graph reference, the CPC graph reference had fewer nodes, edges and diversity, probably owing to only Chinese samples being included, compared with HPRC covering both African and European samples (Supplementary Table 8). However, using the CPC graph reference achieved better alignments than using the HPRC graph reference when aligning the East Asian genomes (Supplementary Fig. 14). By contrast, the HPRC graph performed better in processing African samples (Supplementary Fig. 14). These results indicate that using population-specific graph references improved the alignment quality of short reads.

To carry out variant calling, we mapped the GAM file in the graph reference coordinate to the BAM file in the linear reference coordinate. The results showed that the mapping rate of all samples decreased by an average of 0.58% (0.54–0.61%). We speculated that the advantages of the graph reference would be lost when using the traditional linear reference to carry out calling or record variation because the novel sequences in the graph reference were missing in the linear coordinate.

## Comparison with the HPRC pangenome graph

To investigate the previously unidentified components contributed by the East Asian populations in the CPC pangenome graph, we constructed a merged Minigraph-Cactus graph including all 116 assemblies in CPC and 94 assemblies in HPRC[1] (Methods). We identified 5,850,863 (18.4%) small variants and 34,223 (17.1%) SVs that were found only in the CPC assemblies (Fig. 3c), of which each sample included 170,307 (s.d. = 10,904) small variants and each haplotype carried 543 (s.d. = 39) SVs, and more than half of the CPC-specific variants were singletons or doubletons (Fig. 3d). In both 'easy' and 'difficult' regions of the GRCh38 reference defined in GIAB 3.0 (ref. 30), approximately 39% of the CPC-specific small variants could not be annotated in gnomAD v0.1.8 (ref. 31; Supplementary Table 9), suggesting that the East Asian-specific small variants identified with the long-reads-based methods remain a potent supplement to the current short-read-based genetic resources. We found that 16,898 (49.4%) of the CPC-specific SVs overlapped the nearby regions (100 kb upstream and downstream of the gene coding regions) of 6,426 protein-coding genes, in which 4,344 genes were disrupted by SVs spanning more than 1 kb and had the most frequent functional enrichments related to immunological functions, such as humoral immune response (GO:0002455, OR = 5.11, BH-adjusted $P = 8.50 \times 10^{-14}$; and GO:0006959, OR = 2.91, BH-adjusted

$P = 1.64 \times 10^{-11}$; Supplementary Table 10). These CPC-specific SVs also showed an overrepresentation of the laryngitis-related genes according to the disease ontology annotation (DOID:3437 and DOID:786, OR = 16.66, BH-adjusted $P = 0.007$).

Furthermore, we estimated the location distribution of CPC-specific SVs using a sliding-window-based analysis along the autosomes (Methods). Similar to HPRC-specific SVs and common SVs, most of the CPC-specific SVs were located at the centromeric and telomeric regions of chromosomes (Fig. 3e and Supplementary Fig. 15). We next applied a one-tailed Fisher's exact test between the number of CPC-specific SVs and SVs that were also found in HPRC assemblies in different regions, and found 223 hotspots where CPC-specific SVs were significantly enriched compared with other SVs (FDR-adjusted $P < 0.05$), involving 807 protein-coding genes (Fig. 3e) overrepresenting biological functions such as oxygen transport (GO:0015671, OR = 22.66, BH-adjusted $P = 0.008$; and GO:0005344, OR = 24.91, BH-adjusted $P = 0.001$) and haemoglobin structure (GO:0031838, OR = 28.58, BH-adjusted $P = 0.003$; GO:0005833, OR = 24.21, BH-adjusted $P = 0.003$; and GO:0031720, OR = 33.15, BH-adjusted $P = 0.002$; Supplementary Table 11).

Long-read sequencing technologies and pangenome graph-based analysis methods allow us to explore large and complex SVs that were previously difficult to locate in NGS data, thus providing the genetic basis for association studies of these complex loci with physiological function or disease. We found that some of the CPC-specific enriched SVs mentioned above were closely related to the prevalent diseases in East Asia. A remarkable example is the α-globin gene cluster located near the telomere of the short arm of chromosome 16, including five functional genes and two pseudogenes[32], 5'-zeta–pseudozeta–mu–pseudoalpha-1–alpha-2–alpha-1–theta-3' (Fig. 4a). We identified six major haplotypes based on the copy number variations of α-globin genes (HBA1 or HBA2) and ζ-globin (HBZ or pseudogene HBZP1; Fig. 4b) genes from the pangenome graph (Supplementary Table 12). In addition to a deletion (Z2A1) and duplication (Z2A3) involving a copy number change of α-globin found in both CPC and HPRC, we also identified two CPC-specific large SVs: a 20-kb deletion (Z2A0) involving five globin genes and a 10-kb duplication (Z3A2 and Z3A3) involving ζ-globin genes (Fig. 4c). The long deletion in which both α-globins are lost has been widely reported as the Southeast Asian deletion (--SEA, A0 in our haplotype)[33], and is mainly distributed in southern China and Southeast Asia. As previously reported[34], the heterozygote SEA deletion (A2/A0) as well as the loss of one copy of the α-globin gene (A2/A1) is phenotypically silent. The homozygous loss of one α-globin gene (A1/A1) leads to mild anaemia; losing three copies (A1/A0) leads to haemoglobin H disease, and homozygous SEA deletion leads to severe hydrops fetalis. The precise localization of the complex SVs on the α-globin gene cluster in the CPC pangenome graph could provide a potential reference for future anaemia-related studies. Another example is the RASA4 gene located on chromosome 7 (Fig. 4d). As compared to the two copies of the reference genome, a high diversity of copy numbers in East Asian populations (Supplementary Table 13), including a six-copy variant that is not found in HPRC samples, was discovered (Fig. 4e). CNVs of this gene have not yet been described. The aberrant expression of RAS p21 protein activator 4, encoded by RASA4, has been widely reported to be closely associated with the development of a variety of human cancers[35], and we observed differences in the dosage frequency distribution among populations (Supplementary Tables 14 and 15), which may contribute to the variation of disease incidence.

We next investigated to what extent the novel SVs identified in the CPC assemblies may increase our insights into disease genetics. On the basis of the 243,465 phenotype-associated variants collected from the latest release of the GWAS catalogue, in which 62,393 variants were reported or replicated in the East Asian populations, we found that 75.95% of the novel SVs >1 kb in size (spanning 83.17% of the total novel sequence length) were located <50 kb from the GWAS loci, and in particular, 55.49% (spanning 72.95% of the total novel sequence length)

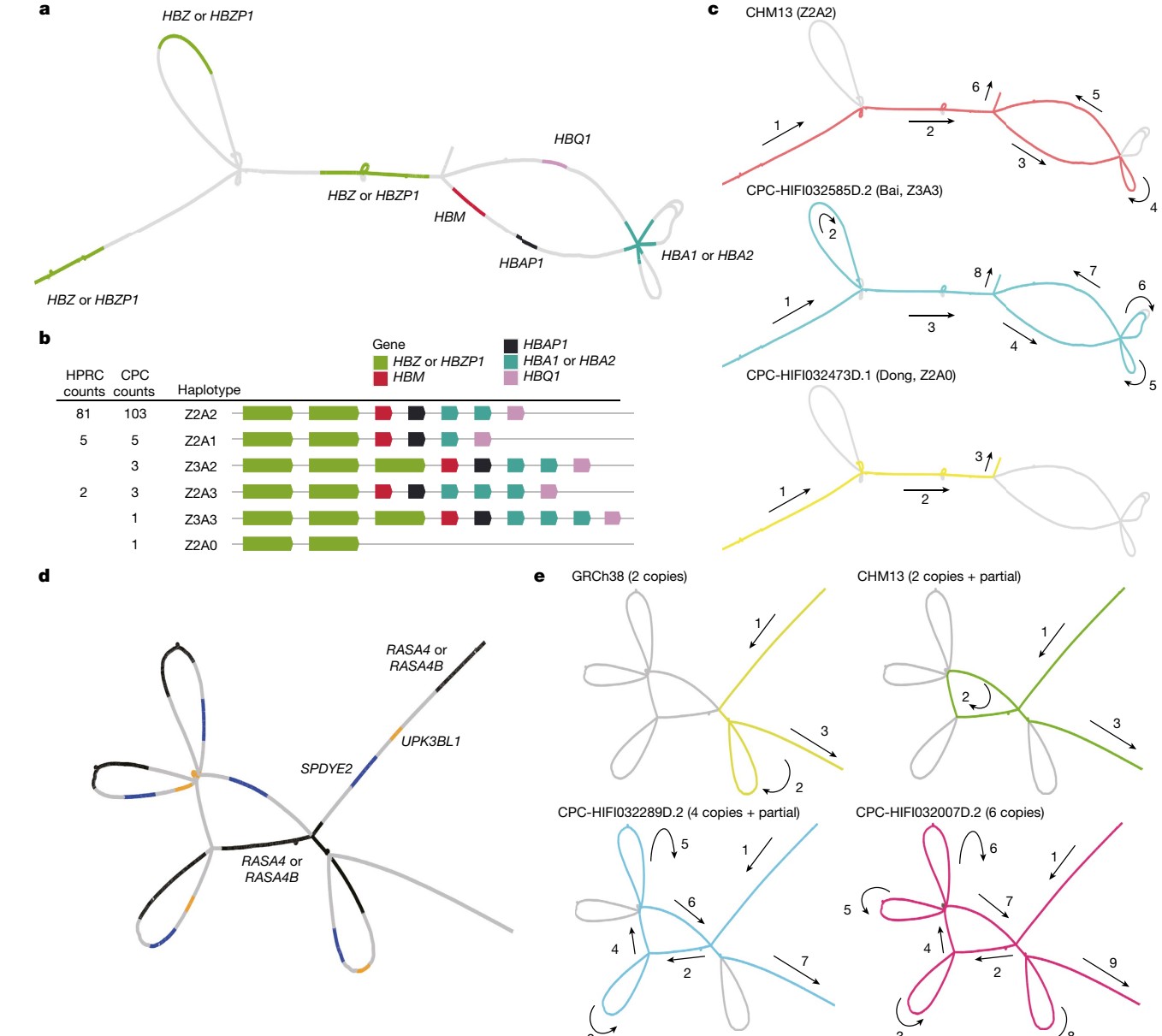

**Fig. 4 | Visualization of novel and complex SVs in the CPC pangenome graph.**
**a**, The locations of α-globin genes on the CPC pangenome subgraph. **b**, Allele counts and linear structural visualization of all structural haplotypes from the Minigraph-Cactus graph among 116 CPC haploid assemblies and 94 HPRC haploid assemblies. The size and spacing of genes on the diagram do not represent the actual size of the chromosome. **c**, Paths of different α-globin gene haplotypes through the joint subgraph. The arrows indicate the direction of the paths. **d**, The locations of genes in the *RASA4* region on the CPC subgraph. **e**, Paths of different structural haplotypes with diverse copy numbers of *RASA4B*. 'partial' represents a 14.9-kb fragment of *RASA4B*.

were around the variants associated with East Asian phenotypes. We observed that, when comparing reported variants across traits, height-associated variants were more likely to be associated with larger proportions of essentially independent novel loci (50.7% and 47.7% for the height-associated variants reported in global and East Asian populations, respectively) than other traits, possibly owing to the remarkable polygenicity of height (Extended Data Fig. 9 and Supplementary Table 16). Moreover, these novel SVs were significantly enriched at disease-associated variants identified in East Asians, including urolithiasis, nephrolithiasis and goitre (BH-adjusted $P$ value = 0.043 for each), which are highly prevalent in some Asian areas[36] (Extended Data Fig. 9).

Despite the fact that all of the novel SVs detected in the CPC assemblies collectively showed a similar level of nucleotide diversity measured by Tajima's $D$ to the rest of the genome, the CPC-specific novel SVs absent in the HPRC assemblies exhibited a significantly higher Tajima's

$D$ than the latter ($P = 4.65 \times 10^{-9}$, one-tailed Wilcoxon rank-sum test; Supplementary Fig. 16a). The most outstanding signals encompassed two protein-coding genes, *STARD7* and *ITPRIPL1*, in chromosome 2 (Tajima's $D = -3.06$, FDR-adjusted $P = 7.54 \times 10^{-12}$; Supplementary Fig. 16b). Again, we highlight the *SPATA31* genes as they could confer evolutionary significance in hominoids not only with the copy number variables (Supplementary Fig. 9) but also with the underrepresented sequence variants (Tajima's $D = -3.01$, FDR-adjusted $P = 2.38 \times 10^{-5}$). All of these results imply a great potential for the novel SVs in the CPC assemblies to provide new insights into the human adaptive evolution in East Asia.

## Archaic introgression and annotation

We applied ArchaicSeeker 2.0 (refs. 37,38) and identified 5,338 archaic introgression segments (AISs) in all 61 CPC samples, spanning

703.87 Mb in total, and on average 84.67 Mb per sample. Of these AISs, 2,450 were located in the coding sequence of 5,531 genes (Supplementary Table 17). We found that 4,126 genes were detected with AISs in at least two samples. In particular, 2,617 genes were detected with AISs in at least five samples, and were substantially enriched in functional categories such as keratinization (GO:0031424, OR = 4.19, BH-adjusted $P = 1.23 \times 10^{-5}$), type I interferon receptor binding (GO:0005132, OR = 130.05, BH-adjusted $P = 7.62 \times 10^{-12}$), positive regulation of peptidyl-serine phosphorylation of STAT protein (GO:0033141, OR = 48.22, BH-adjusted $P = 1.23 \times 10^{-10}$), RIG-I-like receptor signalling pathway (hsa04622, OR = 4.04, BH-adjusted $P = 2.44 \times 10^{-4}$) and neuronal cell body (GO:0043025, OR = 1.75, BH-adjusted $P = 5.96 \times 10^{-3}$; Supplementary Fig. 17 and Supplementary Table 18). We obtained similar results when analysing all AIS-affected genes and those detected with higher-frequency AISs (1,510 genes with AISs carried by at least 10 samples; Supplementary Tables 19 and 20 and Supplementary Figs. 18 and 19). The extremely high-frequency AISs (>40 samples) affected the following genes (annotated with GeneCards Suite[39] online): *KRT6C*, *KRT6A*, *KRT6B* and *KRT75*, which are all keratin gene family members; *CACNA2D2*, *CYB561D2*, *EEF1A2* and *KCNQ2*, which are associated with developmental and epileptic encephalopathy; *GNAT1*, which functions as a signal transducer in normal rod photoreceptor (RHO)-mediated light perception by the retina, and is associated with autosomal recessive congenital stationary night blindness[40]; and *USH2A*, which is involved in cell development and maintenance of the inner ear and retina, and is associated with Usher syndrome and retinitis pigmentosa in the Chinese population[41].

The CPC assemblies are enriched with archaic hominin sequences compared with the African samples in the HPRC dataset (Supplementary Fig. 20). We further compared the AISs detected in the CPC samples and those in the American samples that constitute the largest continental group in the HPRC dataset second only to the African group. We found that the proportion of the Altai Neanderthal-like AISs was higher in the Americans (76.44 ± 15.61 Mb) than in the CPC East Asians (74.39 ± 4.48 Mb) on both the individual and population levels given comparable sample size (525.81 Mb for American, and 434.54 ± 8.42 Mb for East Asian). The Denisovan-like AIS proportion was higher in the East Asian genome (16.92 Mb, covering 0.59% of the American genome, and 2.10 Mb ± 0.71 Mb (0.07%) for each sample; 26.04 ± 1.34 Mb, covering 0.90% on average of the East Asian genome, and 2.77 Mb ± 0.70 Mb (0.10%) for each sample), indicating greater AIS diversity inherited from the Denisovan in the East Asian genomes than in the American genomes (Supplementary Fig. 20). In addition, the archaic hominin introgression in East Asians was largely underrepresented by the CHS samples in HPRC. Each population in the CPC assembly on average added 15.45 Mb of AISs (14.16 Mb of Altai Neanderthal-like sequences and 1.29 Mb of Denisovan-like sequences) to the archaic sequence pool of the present-day East Asians (Supplementary Fig. 21), and each CPC genome contributed 9.56 Mb of archaic-like sequences. In particular, the Turkic-speaking populations (for example, Uyghur, Kazakh and Kyrgyz) showed the least Altai Neanderthal-like AIS sharing with other East Asian populations, possibly owing to the European genetic ancestry in these populations (Extended Data Fig. 10a); some southern Chinese linguistic groups (for example, Tai–Kadai and Austro-Asiatic) added to the Denisovan-like AIS diversity at the highest level (Extended Data Fig. 10b).

We further investigated genes affected by the CPC-specific AISs and their potential functions. We found 1,575 AISs spanning 72.41 Mb in the CPC assembly that were absent in the HPRC assembly. These CPC-specific AISs encompassed 3,629 genes in total. We highlighted 1,211 genes affected by potentially functional AISs located in the coding sequence regions (Supplementary Table 21), which had roles in xenobiotic glucuronidation (GO:0052697, OR = 77.51, BH-adjusted $P = 1.22 \times 10^{-6}$), flavonoid metabolic processes (GO:0009812, OR = 28.73, BH-adjusted $P = 3.98 \times 10^{-6}$) and ascorbate and aldarate metabolism (hsa00053,

OR = 8.66, BH-adjusted $P = 8.88 \times 10^{-4}$; Supplementary Fig. 22 and Supplementary Table 22). According to the GeneAnalytics[42] annotation, these genes are associated with multiple diseases (for example, colorectal cancer, breast cancer, schizophrenia and nervous system disease) (Supplementary Table 23). We found that a CPC-specific AIS affecting *BOD1* was carried by 71 (61.2%) haploid assemblies. This gene was reported to be involved in cerebellar motor dysfunction[43], and is crucial for human cognitive function[44]. Another AIS-affected gene, *IL17RA* ($n_{haplotype} = 52$; 44.8%), has a pathogenic role in many inflammatory and autoimmune diseases. In particular, polymorphism at this gene is related to atopic dermatitis, autoimmune type 1 diabetes and asthma in East Asian populations[45]. *TWIST2* ($n_{haplotype} = 41$, 35.3%) and *CHFR* ($n_{haplotype} = 34$; 29.3%) have critical roles in cancer metastasis, and are commonly used biomarkers for various cancers[46]. All of the population-specific AISs >150 kb in size are listed in Supplementary Table 24, and almost all were present only in one single individual ($n_{haplotype} = 1$). The Uyghur population contributed the largest proportion (0.51%, 14.68 Mb) of the CPC-specific archaic introgression among all of the ethnicities studied (Supplementary Table 25). One notable example of the Uyghur-specific AIS affects *QPCT*, which codes for glutamyl peptidyltransferase. This gene is associated with schizophrenia in both European and Han Chinese populations[47], and may also affect bone mineral density in adult women, resulting in susceptibility to osteoporosis[48]. Moreover, we found a well-recognized oncogene, *JUN*, affected by archaic introgression in the Zhuang population, which could be responsible for the present-day differential prevalence and association with cancers for the *JUN* variants.

We found that 6.68% of the AISs identified in the CPC assemblies were attributed to genes affected by the SVs, and 17.68% of the SVs were affected by archaic introgression (Supplementary Table 26). In addition, 0.10% of these AISs were detected in 141 CNV genes in the CPC assemblies (Supplementary Table 4); in particular, 0.09% were detected in 135 CPC-specific CNV genes. These results imply that the CPC data hold great potential to advance our understanding of human evolutionary history in Asia.

## Discussion

In this study, we sequenced 58 CPC core samples to an average depth of 30.65× using PacBio HiFi long-read sequencing. With an average contiguity N50 > 35.63 Mb and an average total size of 3.01 Gb, the 116 high-quality and haplotype-phased de novo assemblies have good coverage of the Telomere-to-Telomere Consortium haploid assembly T2T-CHM13. Our analysis showed that the CPC assemblies largely matched or exceeded the continuity and base-level accuracy of the current reference human genome sequence (GRCh38). The CPC core assemblies also have good coverage of GRCh38, and added 189 million base pairs of euchromatic polymorphic sequences and 1,367 protein-coding gene duplications to GRCh38. The CPC Phase I data thus serve as a comprehensive pangenome reference for Chinese populations and are expected to better capture genomic diversity in populations of Asian ancestry. Our further analysis confirmed the necessity of high-quality population-specific assemblies for genetic and medical applications[2]. Indeed, we identified variations showing considerable differentiation among different ethnic groups, probably resulting from divergent ancestral backgrounds. Our results also suggest that the use of population-specific references in sequence alignment improved the alignment quality. Compared with the HPRC graph reference, using the CPC graph reference improved the perfect alignment rate of short reads in East Asian samples.

The CPC pangenome reference undoubtedly provides a more comprehensive understanding of genomic variation in Asian populations, particularly those of Chinese ancestry. For example, about 18.4% of the small variants and 17.1% of the SVs identified were specific to the CPC assemblies compared with the HPRC data, although most of the CPC-specific SVs were located at the centromeric and telomeric regions

of chromosomes. More than half of the variants showed an extremely low frequency, such as singletons or doubletons, and they were specifically identified in either CPC or HPRC data. Therefore, our results indicated the necessity of a more comprehensive sampling effort for both CPC and HPRC. Meanwhile, we have also generated a joint CPC–HPRC pangenome reference with both CPC and HPRC assemblies so that it can be more widely applied to various enterprises.

The CPC data also demonstrated a remarkable increase in the discovery of novel sequences when individuals were included from genetically divergent ethnic groups. A notable example is the α-globin gene cluster, in which we identified a 20-kb deletion that has been widely reported as a cause of anaemia specifically in the southern Chinese and Southeast Asian populations, and a 10-kb duplication specific to CPC assemblies. Therefore, our analyses demonstrated great potential in discovering novel or missing sequences in underrepresented Asian populations, especially minority ethnic groups.

Although not surprising, we identified a substantial proportion of sequences of archaic origins. In particular, every ethnic group contributed on average about 15 Mb and every sample contributed about 9.5 Mb of sequences of archaic ancestry, indicating the potential for discovering novel archaic sequences that were missing in previous studies. Moreover, the novel archaic sequences identified in the CPC assemblies were largely underrepresented in the HPRC data, which again suggests the necessity of including more diverse samples of Asian ancestry in further efforts of the HPRC. An interesting observation was that the least Altai Neanderthal-like sequences were shared between the Turkic-speaking populations from northwestern China (for example, Uyghur, Kazakh and Kyrgyz) and other East Asian populations, probably owing to the genetic admixture with west Eurasian populations, which diluted the Altai Neanderthal-like ancestry in northwestern Chinese populations.

We showed previously that individuals of Chinese or Asian ancestry harbour a great genomic diversity[6]. China is populated with multiple ethnic groups with high cultural and language diversities, including 55 officially recognized minority ethnic groups in addition to the Han Chinese majority and a considerable number of unrecognized ethnic groups. As the first effort (Phase I) of the CPC, the current pangenome reference constructed by the CPC was based on 58 CPC core samples representing 36 of the 55 minority ethnic groups and 8 linguistic groups. In its plans, the CPC aims to produce high-quality, phased, chromosome-level haplotype sequences of 500 individuals, which will cover the 56 ethnic groups as officially defined as well as a number of unidentified ethnic groups that have never been well covered by any previous work, such as Sherpa[49], Dolan, Keriyan, Deng and Lop Nur. The fully phased T2T diploid genomes will represent a paradigm shift and the new standard in population-level genomic studies[7]. In parallel with the effort to document genomic diversity, considerable efforts have been invested in comprehensively annotating the elements in the CPC genomes that confer function, such as genes, control elements and transcript isoforms. Annotating the CPC genomes resulted in discovering genes that confer essential functions and underlying natural selection, which are probably associated with phenotypic diversity of disease susceptibility specific to Asian populations. In particular, a considerable proportion of the CPC sequences are of archaic origins and enriched in genes related to keratinization, inflammation and autoimmune diseases. Moreover, the novel sequences specifically discovered in the CPC pangenome encompassing 6,426 protein-coding genes confer phenotypic diversity or disease susceptibility, including immunological functions. Taken together, the CPC Phase I data have already demonstrated a great potential to shed new light on human evolution and recover missing heritability in complex trait and disease mapping. We expect the CPC, as an important part of the global force of human genomics, to make a considerable contribution to building high-quality pangenome references and applying them for various basic and clinical research projects.

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

**Chinese Pangenome Consortium (CPC)**

**Chuangxue Mao[3], Shaohua Fan[1], Qiang Gao[17], Juncheng Dai[13,14], Fengxiao Bu[19], Guanglin He[19], Yang Wu[19], Huijun Yuan[19], Jinchen Li[20,21], Chao Chen[21], Jian Yang[22,23], Chaochun Wei[24], Xin Jin[25] & Xia Shen[1,26]**

[19]Institute of Rare Diseases, West China Hospital of Sichuan University, Sichuan University, Chengdu, China. [20]Bioinformatics Center & National Clinical Research Centre for Geriatric Disorders, Xiangya Hospital, Central South University, Changsha, China. [21]Center for Medical Genetics & Hunan Key Laboratory of Medical Genetics, School of Life Sciences, and Department of Psychiatry, The Second Xiangya Hospital, Central South University, Changsha, China. [22]School of Life Sciences, Westlake University, Hangzhou, China. [23]Westlake Laboratory of Life Sciences and Biomedicine, Hangzhou, China. [24]Department of Bioinformatics and Biostatistics, School of Life Sciences and Biotechnology, Shanghai Jiao Tong University, Shanghai, China. [25]BGI-Research, Shenzhen, China. [26]Center for Intelligent Medicine Research, Greater Bay Area Institute of Precision Medicine (Guangzhou), Fudan University, Guangzhou, China.

## Methods

### Populations and samples

For Phase I of the CPC project, we selected 68 samples from 731 individuals with genomes deep-sequenced using next-generation sequencing. Following a previous study[5], we applied a procedure to quantitatively evaluate the genetic diversity coverage based on principal component analysis results. We selected individuals using a statistic $D_d$ to measure the representation of population samples, and found that the selected samples are located close to the centre of the cluster of each population on the plot of the top two principal components. In addition, we strived to maintain the balance of sex ratio in the sample selection of each population. After quality control of the HiFi data, eventually, 58 samples were retained for all of the downstream analyses in this study.

### Ethics and inclusion

The CPC is a joint effort of scientists of various minority ethnic groups who represent underrepresented populations and minority groups. From the beginning, the CPC has involved many local scientists of various minority ethnic groups. The design and execution of the CPC have included inputs from local scientists from the beginning and throughout the entire process. As our co-authors, they have substantially contributed research work, they are well educated and aware of the ethical issues, and they are capable of explaining explicitly the results to their local minority communities. Informed consent was obtained from all individual participants included in the study. The personal identifiers of all samples, if any existed, were stripped off before sequencing and analysis. All procedures were carried out in accordance with the ethical standards of the Responsible Committee on Human Experimentation and the 1964 Helsinki Declaration, its later amendments (2000), or comparable ethical standards. The research content and procedures carried out in studies involving human participants were approved by the Biomedical Research Ethics Committee of Shanghai Institutes for Biological Sciences (number ER-SIBS-261408), the Biomedical Research Ethics Committee of Kunming Institute of Zoology, Chinese Academy of Sciences (number SMKX-20180715-154) and the Biomedical Research Ethics Committee of The First Affiliated Hospital of Xi'an Jiaotong University (number XJTU1AF2021LSK-051).

### DNA sample and library preparation

Genomic DNA was extracted from peripheral blood samples and Epstein–Barr virus-immortalized B lymphoblastoid cell lines using the optimized cetyl trimethyl ammonium bromide-based protocol followed by purification with QIAGEN Genomic kit (catalogue number 13343, QIAGEN) for regular sequencing, according to the standard operating procedure provided by the manufacturer (Supplementary Information). These lymphoblastoid cell lines are limited in their growth to minimize genetic changes that may occur during extended cell culture. DNA degradation and contamination were monitored on 1% agarose gels. DNA purity was checked using the NanoPhotometer spectrophotometer (IMPLEN). DNA concentration was measured using the Qubit DNA Assay Kit in a Qubit 2.0 fluorometer (LifeTechnologies).

The library of 15 kb was constructed using a SMRTbell Express Template Prep Kit 2.0 (Pacific Biosciences). The construction includes DNA shearing, damage repair, end repair, hairpin adapter ligation, size selection and purification of the library (Supplementary Information).

### Hi-C sequencing

To anchor hybrid scaffolds onto the chromosome, genomic DNA was extracted for the Hi-C library from blood. We followed the standard protocol described previously[50,51] with certain modifications. In brief, we constructed the Hi-C library and obtained sequencing data through the Illumina Novaseq-6000 platform. The ligated DNA was sheared into 300–500-bp fragments, and then was blunt-end-repaired and A-tailed, followed by purification through biotin–streptavidin-mediated pull-down. Finally, the Hi-C libraries were quantified and sequenced using the Illumina Novaseq-6000 platform.

### NGS

A total amount of 1.5 μg DNA per sample was used as input material for the DNA sample preparations. Sequencing libraries were generated using the Truseq Nano DNA HT Sample Preparation Kit (Illumina) following the manufacturer's recommendations and index codes were added to attribute sequences to each sample. In brief, the DNA sample was fragmented by sonication to a size of 350 bp, and then DNA fragments were end polished, A-tailed and ligated with the full-length adapter for Illumina sequencing with further PCR amplification. Finally, PCR products were purified (AMPure XP system), and libraries were analysed for size distribution by an Agilent 2100 Bioanalyzer and quantified using real-time PCR. These libraries constructed above were sequenced by the Illumina NovaSeq-6000 platform and 150-bp paired-end reads were generated with an insert size of around 350 bp.

### HiFi sequencing

SMRTbell target size libraries were constructed for sequencing according to PacBio's standard protocol (Pacific Biosciences) using 15-kb preparation solutions. In brief, a total amount of 15 μg DNA per sample was used for the DNA library preparations. The genomic DNA sample was sheared by g-TUBEs (Covaris) according to the expected size of the fragments for the library. Single-strand overhangs were then removed, and DNA fragments were damage-repaired, end-repaired and A-tailed. Then the fragments were ligated with the hairpin adapter for PacBio sequencing. Then the library was treated by nuclease with SMRTbell Enzyme Cleanup Kit and purified by AMPure PB beads. Target fragments were screened by BluePippin (Sage Science). The SMRTbell library was then purified by AMPure PB beads, and an Agilent 2100 Bioanalyzer (Agilent Technologies) was used to detect the size of library fragments. The library was sequenced using a single 8M SMRT Cell on the PacBio Sequel II or Sequel IIe platform (Pacific Biosciences) with Sequencing Primer V2 and Sequel II Binding Kit 2.0 in Grandomics. The PacBio SMRT-Analysis package (https://www.pacb.com) was used for the quality control of the raw polymerase reads (Supplementary Information).

### Oxford Nanopore Technologies sequencing

A total amount of 3–4 μg DNA per sample was used as input material for the Oxford Nanopore Technologies library preparations. After the sample was qualified, size selection (>50,000 and >100,000) of long DNA fragments was carried out using the PippinHT system (Sage Science). Next, the ends of DNA fragments were repaired, and A-ligation reactions were conducted with NEBNext Ultra II End Repair/dA-tailing Kit (catalogue number E7546). The adapter in SQK-LSK109 (Oxford Nanopore Technologies) was used for further ligation reaction and the DNA library was measured by Qubit 4.0 Fluorometer (Invitrogen).

### Genome assembly and quality control

A total of 68 samples were sequenced on 2–5 SMRT Cells. The obtained subreads are converted into HiFi reads by ccs v6.3.0 in PacBio tools with --hifi-kinetics --min-passes 3 --min-length 50. At the individual level, we combined HiFi reads from multiple cells and applied Hifiasm v0.16.1 (ref. 52) to carry out both the primary assembly and diploid assembly for 68 PacBio HiFi samples (Supplementary Information).

We used QUAST v5.2.0 (ref. 53) to assess the assembly quality of 68 primary assemblies and 136 haplotype assemblies (Supplementary Information). After quality control, we removed three samples with primary assembly N50 < 20 Mb or contig number ≥ 2,000, and seven samples with two haplotype assembly N50 < 10 Mb or contig number ≥ 2,000. Finally, 58 samples with 116 high-quality assemblies were retained for subsequent analyses.

## Assembly polishing and assessment

We first used Inspector v1.2 (ref. 54) to evaluate the assembly errors of 116 assemblies, which were recorded as bed files in the output directory of Inspector. We then carried out assembly polishing based on the evaluation results of assembly errors of Inspector using the following command: inspector-correct.py -i $asm/ --datatype pacbio-hifi -o $asm. corrected/ --skip_structural -t 64. We next applied Inspector to assesses the misassemblies before and after correction to determine whether the assembly polishing improved the assembly quality. The assembly quality value was estimated using Inspector on the basis of the calculation of assembly errors in each assembled haplotype. The assessment command of assembly errors using Inspector was as follows: inspector.py -c $asm.fa -r $sample.ccs.fastq.gz -o $asm/ --datatype hifi -t 64.

We also applied QUAST to assess the contiguity, completeness and duplication ratio for each corrected assembly. The assessment command of QUAST was as follows: quast-lg.py $asm.fa -r $reference.fa -g $reference.gff.gz -o $asm/ --large --est-ref-size 3100000000 --no-icarus -t 64.

## Variant identification from the phased assembly

The obtained HiFi reads were aligned to the T2T-CHM13 v2.0 reference by minimap2 using the preset parameters -ax map-hifi, and then sorted by samtools sort. The quality of reads at the alignment level was evaluated by samtools stats. The BAM file was used for the identification of conventional and complex structural variations (Supplementary Information).

We also applied minimap2 v2.24 (ref. 55) to align each assembly to the T2T-CHM13 v2.0 reference using the following command: minimap2 -x $asm.fa $reference.fa -m 10000 -z 10000,50 -r 50000 --end-bonus=100 --secondary=no -a -t 20 -eqx -Y -O 5,56 -E 4,1 -B 5. We used dna-brnn v0.1 (ref. 56) to identify highly repetitive sequences of each CPC assembly. We then used Phased Assembly Variant (PAV) caller v1.2 (ref. 17), which can prune duplicated alignments caused by minimap2, to detect small variants and SVs in each haplotype on the basis of the output CIGAR sequence from minimap2. We specified the reference genome (T2T-CHM13), two-phased assemblies of each sample and a configuration file as inputs to launch PAV. In our PAV pipeline, SVs within 200 bp with 50% overlap between two assemblies of each sample were merged.

To identify putative novel sequences of CPC samples relative to the T2T-CHM13 reference, we first applied SV-pop v3.0 (ref. 17) to merge SVs of core CPC samples using the sampleset merge and the size-reciprocal-overlap mode with nr::szro(0.8,4):match(0.8) parameters. Under this condition, insertions with at least 80% overlap were merged when the distance between insertions was less than four times the insertion length. We defined the insertions as the novel sequences larger than 1 kb relative to T2T-CHM13 in each SV calling result. We applied primatR v0.1.0 (https://github.com/daewoooo/primatR) to detect the hotspots of novel sequences using the function hotspotter and set the parameters bw = 20,000 and num.trial = 1,000.

## Identification of gene duplication

We used liftoff v1.6.3 (ref. 57) to detect duplicated genes relative to GRCh38 references in each assembly. GENCODE v38 (ref. 58) annotation was used to build the annotation database for liftoff. Protein-coding genes with extra gene copies annotated by liftoff were identified as duplicated genes for each assembly. We ran liftoff with an identity threshold of at least 90% using the parameter -sc 0.9, and the command was as follows: liftoff -p 64 -sc 0.9 -copies -db GENCODE_V38.db -u $asm.unmapped -o asm.gff3 -polish asm.fa GRCh38.fa. We identified gene duplications with the field extra_copy_number in the output gff3 file of liftoff.

## Pangenome graph construction

We applied the Minigraph-Cactus pipeline[59] to construct the CPC pangenome variation graph in two versions: GRCh38 based and CHM13 based. We first constructed the graph with only SVs larger than 50 bp using Minigraph (v0.19)[60]. Starting from the reference assembly, either GRCh38 or CHM13, as the initial graph, the remaining reference genome and 122 assemblies were sequentially mapped to the graph. We used the Minigraph-Cactus pipeline to add SNP-level variants to the graph. First, centromere and telomere regions were identified and softmasked with dna-brnn[56] to exclude the influence of highly repetitive sequence on subsequent analysis. The assemblies were remapped to Minigraph to produce exact alignments between the contigs of the input assembly and Minigraph node sequences. Quality control was applied in this step by excluding all softmasked sequences longer than 100 kb and all mappings with MAPQ < 5. Next, we split the graph and assigned the contigs into different chromosomes, and applied Cactus v2.1.1 base alignment[61] separately, and the output HAL[62] files were converted to the vg format with hal2vg (https://github.com/ComparativeGenomicsToolkit/hal2vg). Paths larger than 10 kb that did not align to the underlying graph were removed, and GFAffix (https://github.com/marschall-lab/GFAffix) was used to normalize the graph. Finally, the graphs of chromosomes were combined into a whole-genome graph, indexed and exported to VCF with the vg toolkit v1.42 (https://github.com/vgteam/vg), and allele filtering was also applied for short-read mapping by removing all nodes traversed by fewer than 12 haplotype paths (minimum AF = 10%) using the command: vg clip -d 12 -m 10000.

## Identification and visualization of CPC-specific variants

We constructed a new Minigraph-Cactus pangenome graph including all 116 assemblies in CPC and 94 assemblies in HPRC with T2T-CHM13.v2 as the reference genome, and the details were consistent with descriptions above. We grouped the alternative alleles in multiallelic SV sites by their length and split them into biallelic records using bcftools norm -m -any. Then we classified the SVs that were found only in the CPC and HPRC assemblies as CPC- or HPRC-specific variants, respectively. We applied RIdeogram v0.2.2 (ref. 63) to visualize the SV distribution and enrichment significance on chromosomes. We extracted the subgraphs of SVs with gfabase v0.6.0 (https://github.com/mlin/gfabase) and then visualized the subgraphs with bandage v0.9.0 (ref. 64). Gene positions in the subgraphs were determined by aligning the sequences of genes to graph with GraphAligner v1.0.16 (ref. 65). Linear structural visualization of all structural haplotypes was applied with the R package gggenes v0.3.1 (https://github.com/wilkox/gggenes).

## Evaluation of the alignment quality of short reads using the CPC reference

To simplify the CPC graph reference, we carried out gradient filtering by removing all nodes traversed by fewer than 6, 9, 12, 15 and 20 haplotype paths using vg clip -d $N -m 10000.

We next used vg stats -zNElL CPC.p1.$N.xg > CPC.p1.$N.stats to estimate the number of nodes and edges in the graph and the total length of the sequence. In addition, we estimated the complexity of the graph by calculating the ratio of edges to nodes (edges/nodes).

We selected ten samples from five East Asian populations of the 1000 Genome Project according to the standard of one male and one female sample per population, and two samples from the African (YRI) population. Fastq files were downloaded from the European Nucleotide Archive (study access: PRJEB31736). We first aligned the paired-end reads of each sample to the graph genomes of the CPC reference with different filter parameters and HPRC by vg giraffe -g CPC.P1.gg -H CPC. P1.gbwt -m CPC.P1.min -d CPC.P1.dist -x CPC.P1.xg -t 128 -f samples1_combined_R1.fastq.gz -f samples1_combined_R2.fastq.gz > sample1. gam. The statistics of the GAM file were obtained through vg stats -a sample1.gam > sample1.qc.

In addition, we also converted the GAM file to linear space (BAM, CHM13) using vg surject -i -x CPC.P1.xg -P -t 128 -b sample1.gam > sample1.bam. The statistics of the BAM files were obtained using samtools stats -@ 16 sample1.bam > sample1.bam.stats.

## Reporting summary

Further information on research design is available in the Nature Portfolio Reporting Summary linked to this article.

## Data availability

The release of the CPC Phase I data by this work is permitted by The Ministry of Science and Technology of the People's Republic of China (permission number 2022BAT2392) at the National Genomics Data Center (https://ngdc.cncb.ac.cn) under the BioProject PRJCA011422. The pangenome references built on the basis of the CPC core samples and combined with the HPRC samples are freely available at both the CPC website (https://pog.fudan.edu.cn/cpc/#/data) and GitHub (https://github.com/Shuhua-Group/Chinese-Pangenome-Consortium-Phase-I).

## Code availability

The code to reproduce the pangenome from this work can be found at GitHub (https://github.com/Shuhua-Group/Chinese-Pangenome-Consortium-Phase-I). Relevant commands used in other analyses can be found in the Methods or Supplementary Information.

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

**Acknowledgements** We thank all of the participants of this project. This study was supported by the National Natural Science Foundation of China (grants 32030020, 32288101, 32125009, 32070663, 31900418, 32270665, 62172325, 31961130380, 82170805 and 81970679), the National Key R&D Program of China (2022YFC3400300 and 2018YFC1311500), the Yunnan Leading Medical Scientist Training Program (L-2018003), the Yunnan Key Research and Development Program (202103AF140002), the Strategic Priority Research Program (XDB38000000) of the Chinese Academy of Sciences, the UK Royal Society-Newton Advanced Fellowship (NAF\R1\191094) and the Shanghai Municipal Science and Technology Major Project (2017SHZDZX01). We also acknowledge the support of the SANOFI Scholarship Program. We thank LetPub (www.letpub.com) for its linguistic assistance during the preparation of this manuscript. The funders had no role in the study design, data collection, analysis, decision to publish or preparation of the manuscript.

**Author contributions** S.X., K.Y. and J.C. conceived and designed the study. S.X. and Y.L. coordinated and supervised the project. Y.L., C.M., Zhaoqing Yang, H.S., J.C., L.J., D.W., J.L., J.D., K.Y., X.Y., B.S., C.C., J.Y., S.W., Z.H. and Xingming Zhao were responsible for collection of blood samples and cell lines. Y.L., Y.G., Zhaoqing Yang, K.Y. and S.X. were responsible for sample selection. Y.L., D.W. and J.D. were responsible for sequencing. Y.L. and L.D. were laboratory organizers within the CPC. L.D., Zhaoqing Yang and J.C were responsible for ethical, legal and social implications. Y.L., Y.G., X.Y., H.C. and Y.C. were responsible for pangenome empirical analysis and pangenome quality control. S.X., Y.L., Y.G., X.Y., H.C., X.T., L.D. and B.W. were responsible for manuscript writing. S.X., Y.L., L.D. and K.Y. were responsible for manuscript editing. H.C., Y.G. and X.Y. were responsible for assembly creation. Y.G., H.C., X.Y., S.L. and Ziyi Yang were responsible for assembly quality control and assembly reliability analysis. Y.G., S.M. and Y.Wang were responsible for pangenome applications of SVs. X.T., X.Y. and S.L. were responsible for pangenome graph creation. B.W. and Y.G. were responsible for data coordination and management. Y.G., H.C. and Y.Wang were responsible for pangenome applications. X.T., X.Y. and S.L. were responsible for pangenome visualization and complex loci analysis. H.C., Y.P., Y.G. and Y.L. were responsible for population genetic analysis. B.W. and C.L. were responsible for archaic ancestry analysis. H.C., X.Y. and S.K. were responsible for variant identification from assembly. L.D., Y.L. and B.W. were responsible for variant annotation. Xiaohan Zhao and Y.S. were responsible for CPC website building.

**Competing interests** The authors declare no competing interests.

**Additional information**
**Correspondence and requests for materials** should be addressed to Yan Lu, Jiayou Chu, Kai Ye or Shuhua Xu.

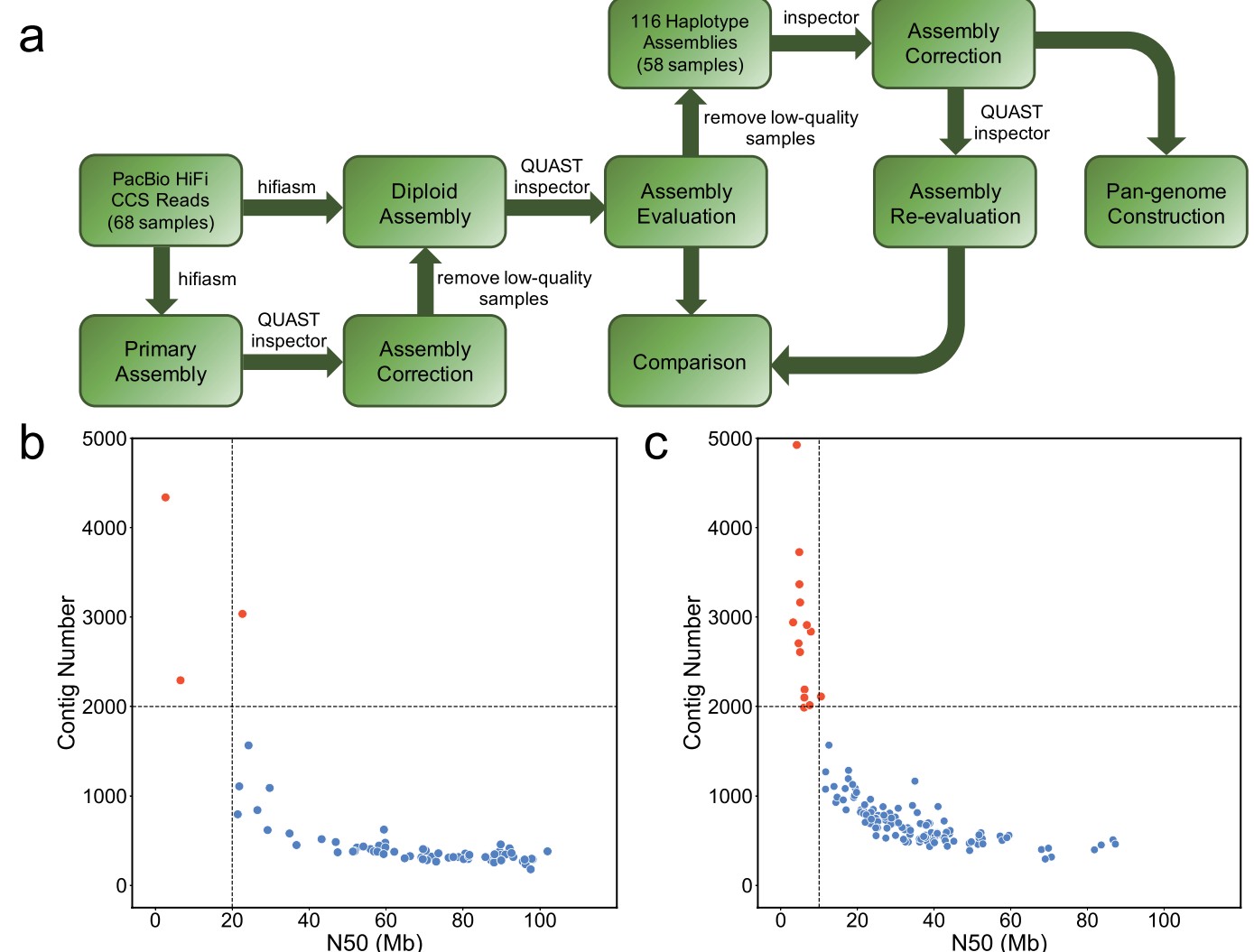

**Extended Data Fig. 1 | Assembly pipeline and quality control of the CPC samples. a**, Flowchart showing the steps and bioinformatic tools applied in quality control, assembly, and correction of 68 CPC samples used for the pan-genome construction. **b**, Quality control of the primary assemblies of 68 CPC samples, in which 3 samples (denoted by the red dots) with N50 <20 Mb or contig number ≥2000 were removed in subsequent analyses. **c**, Quality control of the diploid assemblies of 65 CPC samples, in which 14 haplotype assemblies (denoted by the red dots) of 7 samples with N50 <10 Mb or contig number ≥2000 were removed in subsequent analyses.

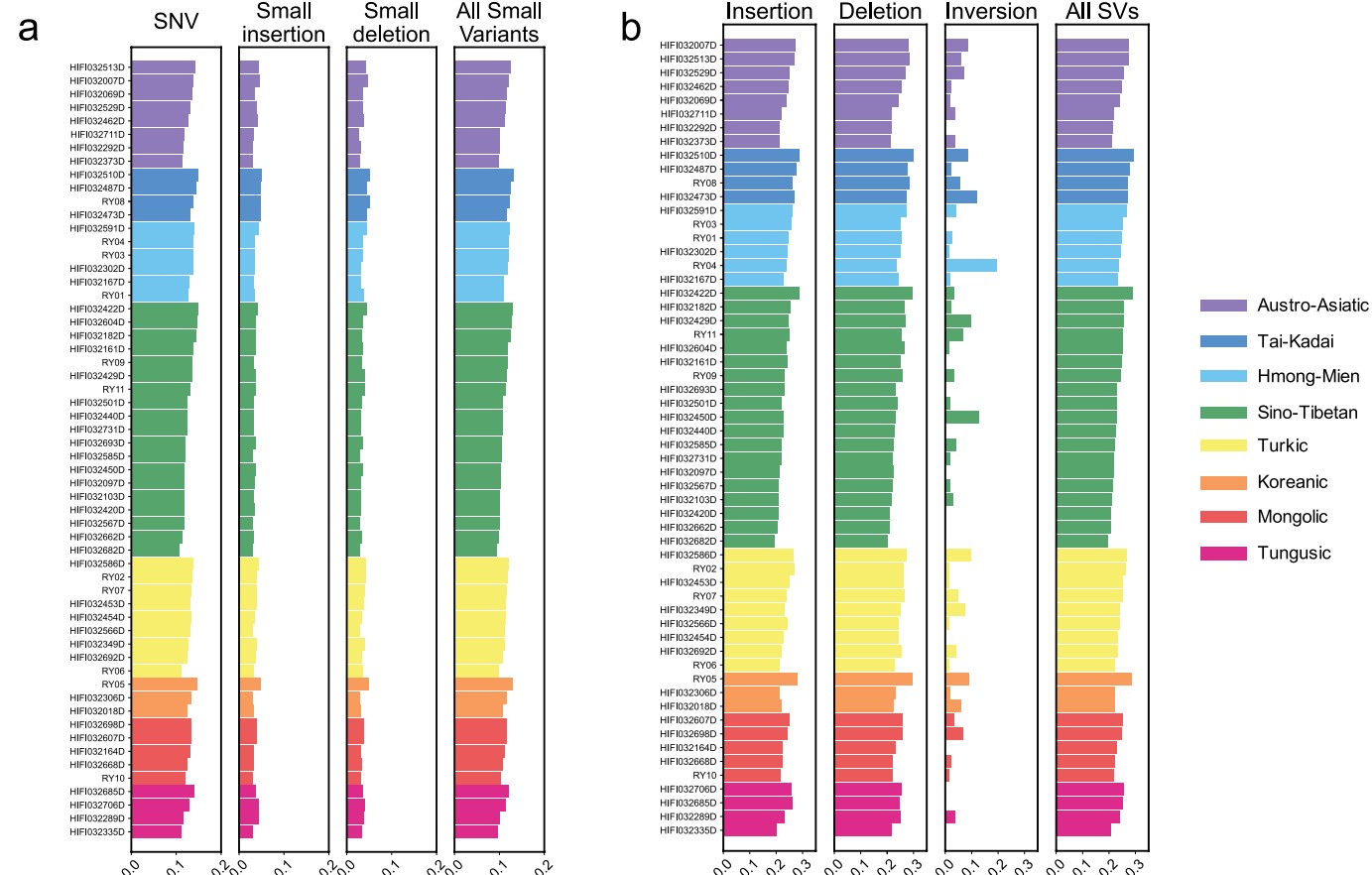

**Extended Data Fig. 2 | Proportion of variant types in the highly repetitive sequences of 58 CPC samples.** The highly repetitive sequences in each assembly were identified using dna-brnn. Proportions of (**a**) small variants and (**b**) SVs in these regions are shown for each sample. In each plot, the samples are ordered by the total variant proportion within each language family.

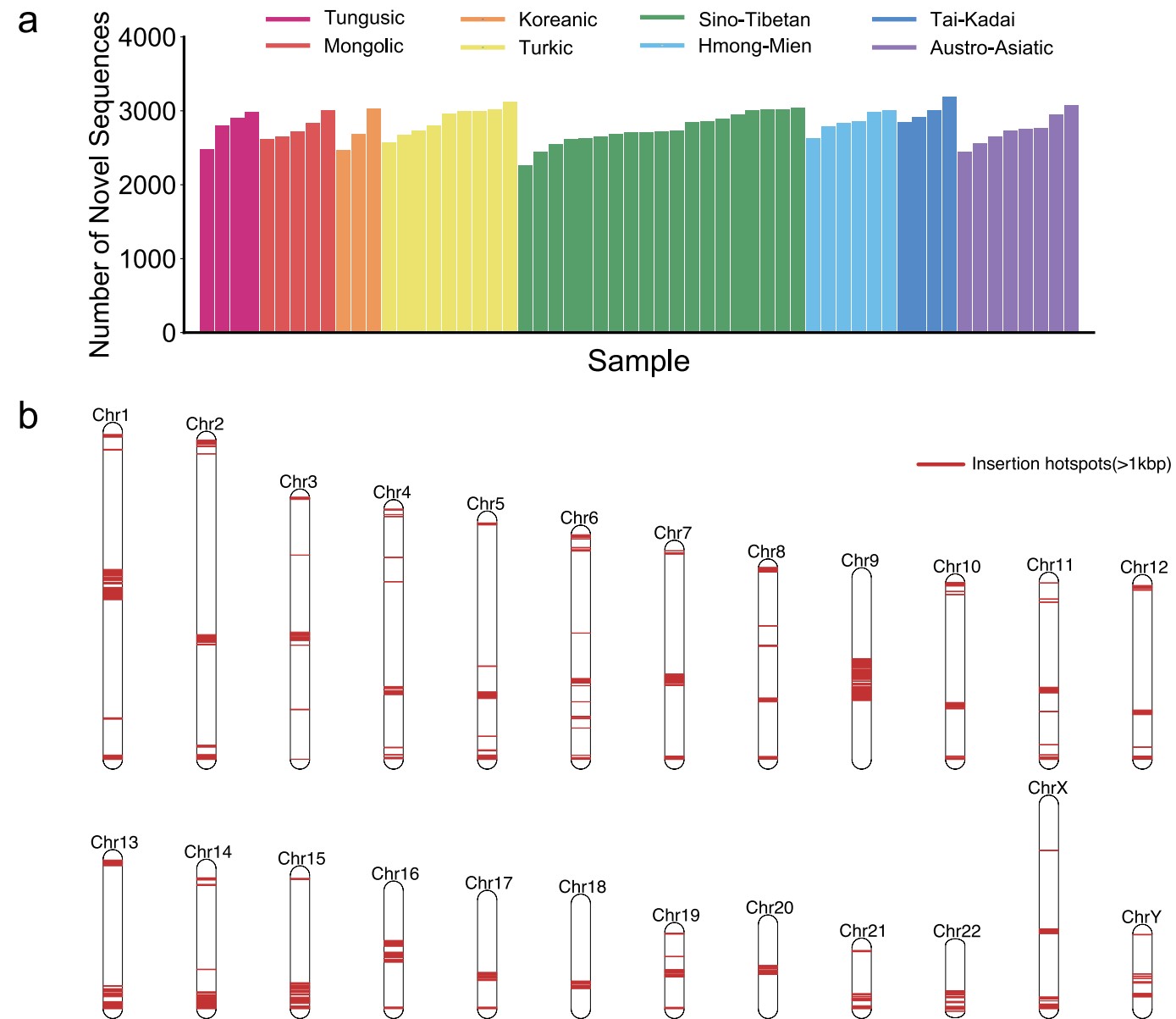

**Extended Data Fig. 3 | Novel sequences with respect to T2T-CHM13 identified in the CPC assemblies. a**, Number of novel sequences (insertions ≥1 kb) identified by aligning two-phased assemblies of each CPC sample to the T2T-CHM13 reference. **b**, Chromosome distribution of 115 insertion hotspots of novel sequences.

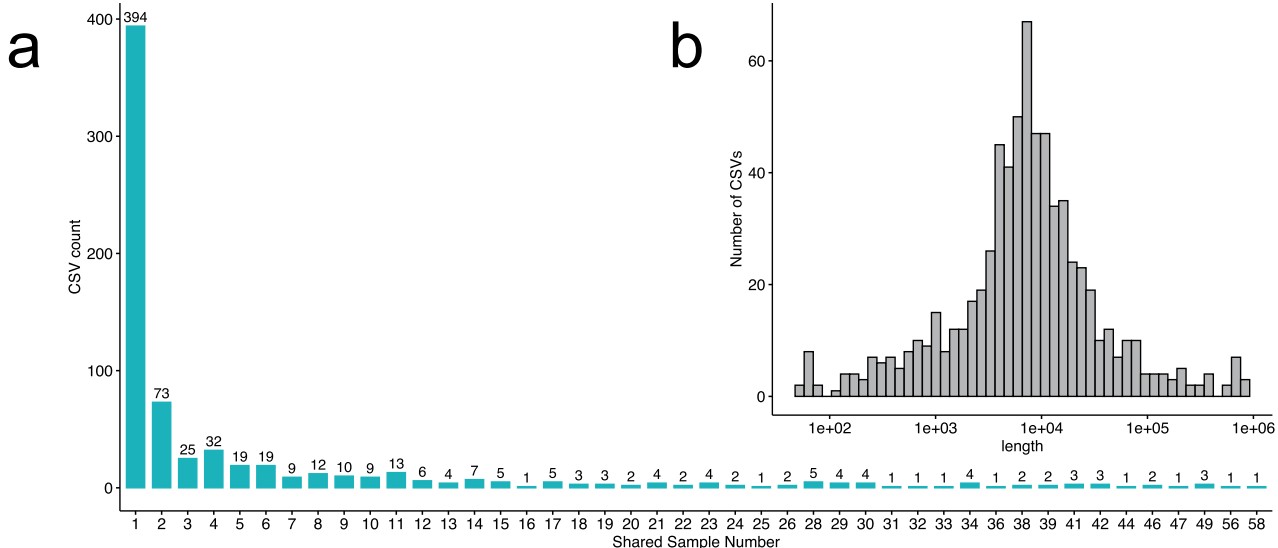

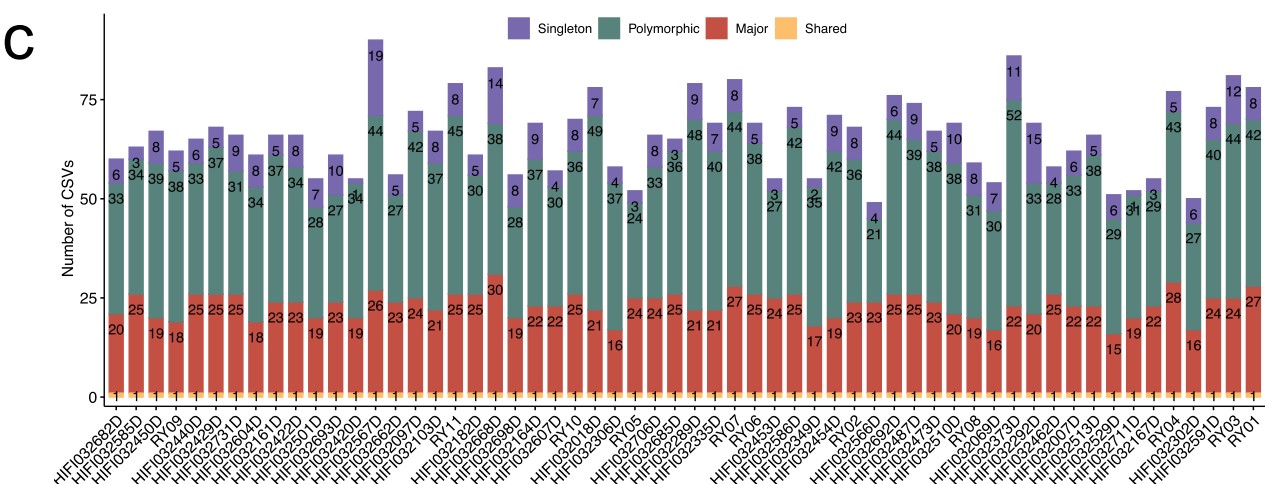

**Extended Data Fig. 4 | CSVs in CPC samples. a**, The count of CSVs with different shared sample numbers. **b**, The length distribution of the final 706 CSVs. We merged CSVs of all 58 samples by considering 80% reciprocal overlap and the same type, and obtained 706 CSVs in total. **c**, The number of CSVs in each CSV class. We classified the CSVs by the shared sample numbers into Shared (present in all samples), Major (present in ≥50% samples but not all), Polymorphic (present in more than 1 but <50% of all samples), and Singleton (present in only one sample).

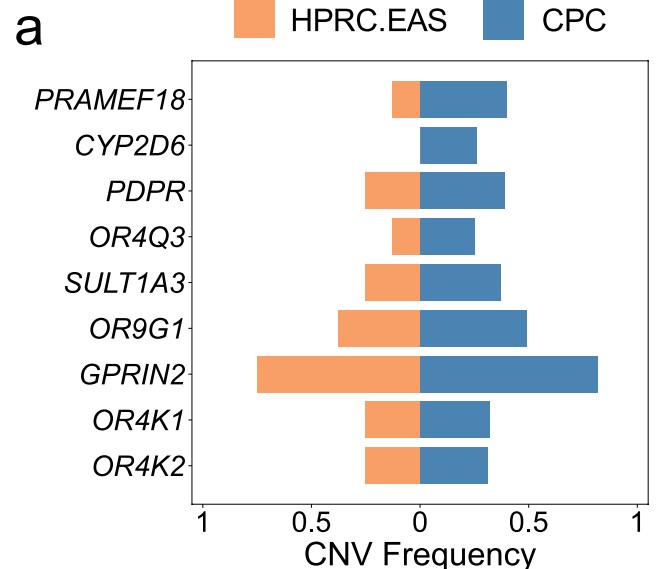
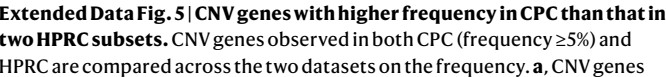
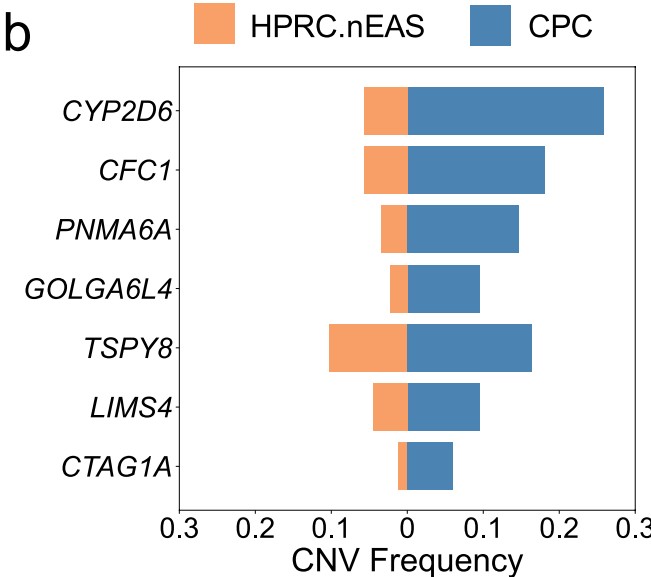

**Extended Data Fig. 5 | CNV genes with higher frequency in CPC than that in two HPRC subsets.** CNV genes observed in both CPC (frequency ≥5%) and HPRC are compared across the two datasets on the frequency. **a**, CNV genes showing higher frequency in the CPC assemblies than in the HPRC.EAS. **b**, CNV genes showing higher frequency in the CPC assemblies than in the HPRC.nEAS from HPRC.

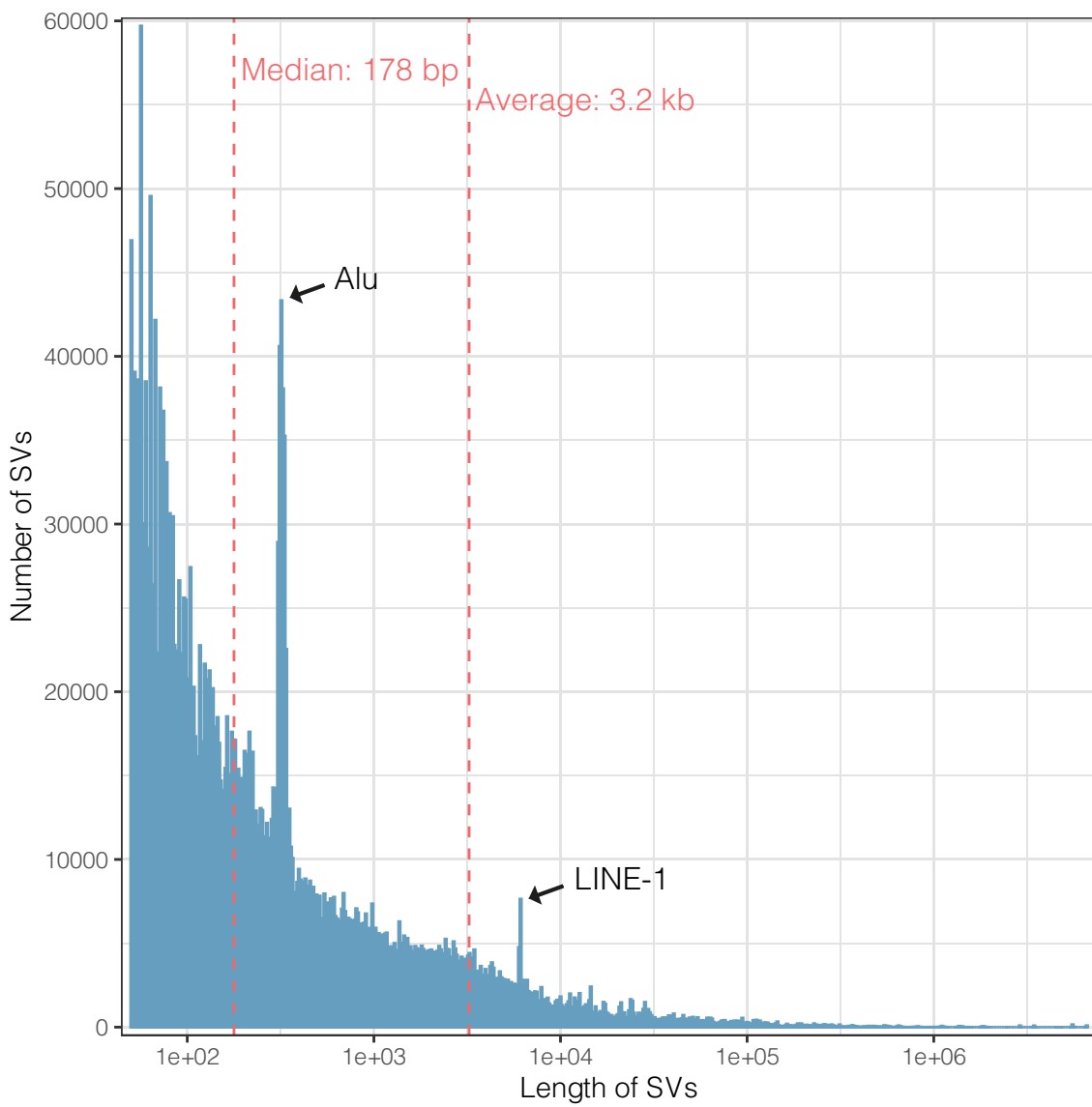

**Extended Data Fig. 6 | Length distribution of SVs in the graph-based pan-genome reference.** The average and median length of SVs are indicated by the two red vertical lines, and both are larger than the read length (usually 150 bp) generated by short-read sequencing. The peaks at 300 bp for Alu insertions and 6 kb for LINE-1 are highlighted.

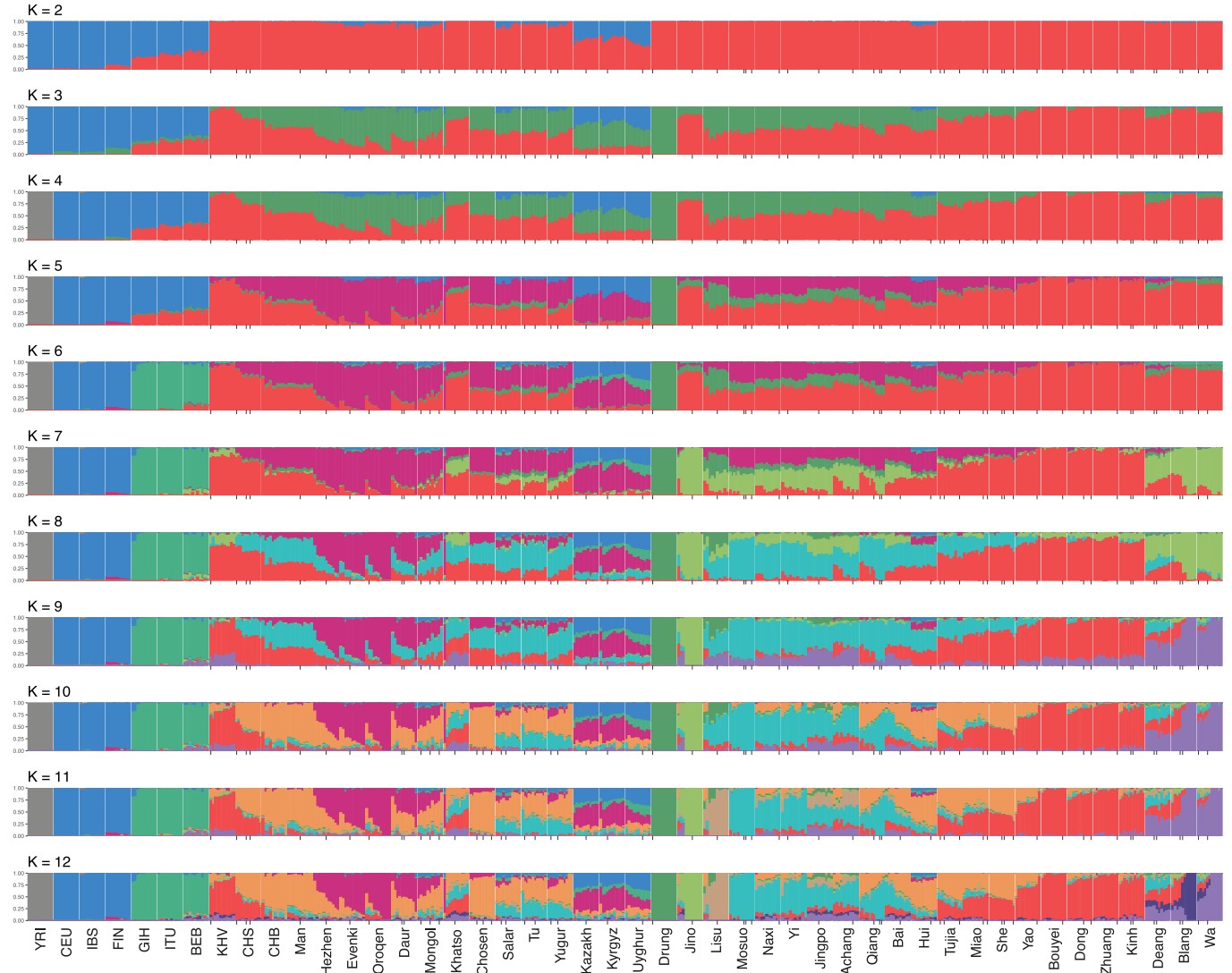

**Extended Data Fig. 7 | Ancestry component inference of the CPC cohort.**
We randomly selected 10 unrelated high-quality samples from each CPC population, and inferred the genetic ancestry for each sample using ADMIXTURE assuming 2–12 ancestry components (K). Samples used in the pan-genome construction are labeled with short vertical lines.

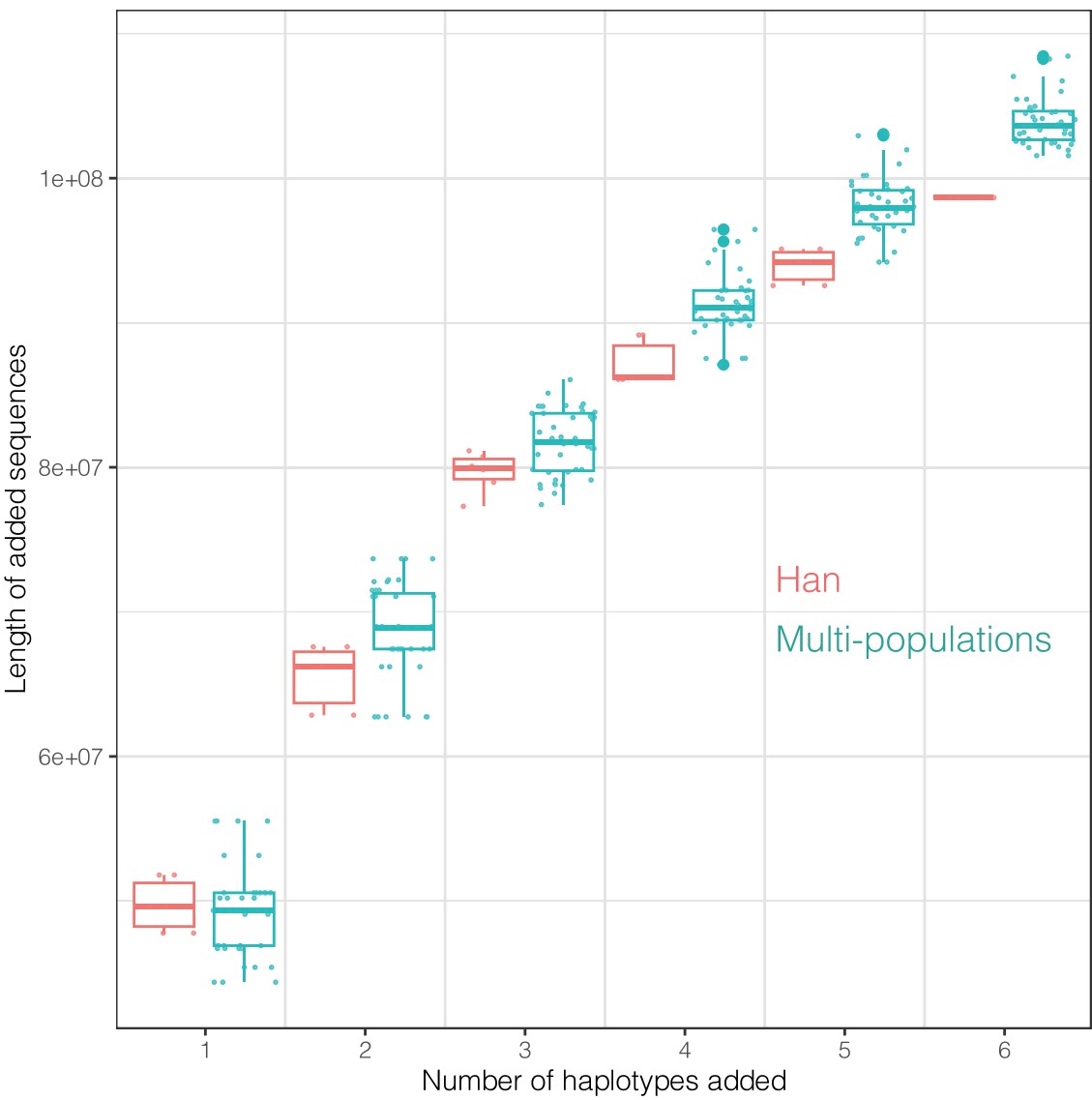

**Extended Data Fig. 8 | Growth of non-reference sequences in Han Chinese and multiple populations.** The cumulative length of non-reference (unaligned to the human reference genome GRCh38) sequences in Han Chinese haplotypes (the red boxes) and that in multi-ethnic haplotypes (the green boxes) are calculated based on the CPC graph genome by pangenome-growth software. We randomly ordered the three Han Chinese samples, and conducted 40 replication analyses with randomly selected samples (three samples for each analysis) from the CPC dataset. The multi-ethnic haplotypes show a relatively larger growth rate of the non-reference sequences than the Han Chinese samples. The lower and upper hinges correspond to the first and third quartiles (the 25th and 75th percentiles). The upper whisker extends from the hinge to the largest value no further than 1.5 * IQR from the hinge (where IQR is the inter-quartile range, or distance between the first and third quartiles). The lower whisker extends from the hinge to the smallest value at most 1.5 * IQR of the hinge. Data beyond the end of the whiskers are called "outlying" points and are plotted individually.

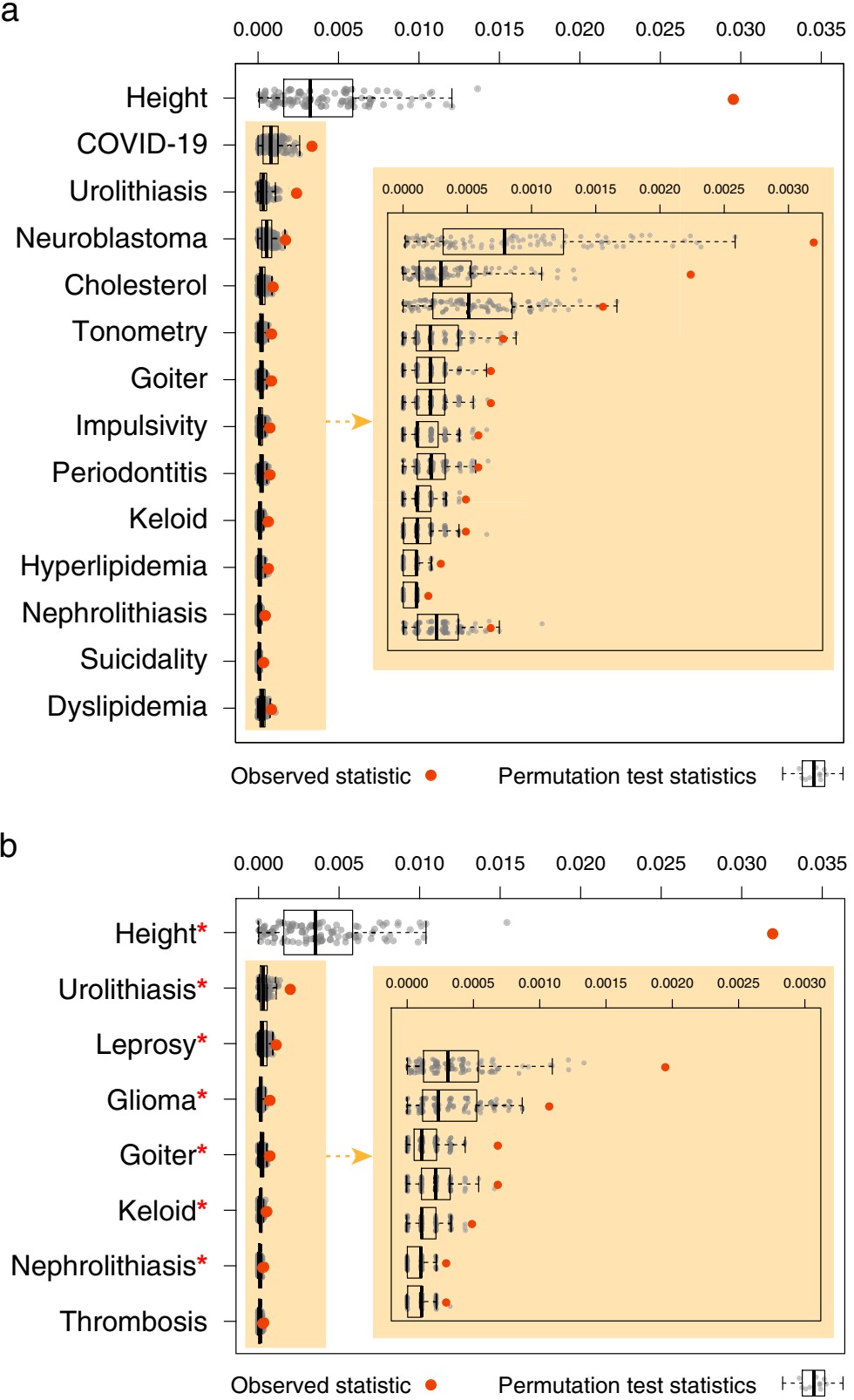

**Extended Data Fig. 9 | Significant enrichment of the CPC-specific SVs identified in the CPC assemblies on the reported GWAS loci.** The proportion of the approximately independent CPC-specific loci (adjacent novel SVs with distance <50 kb are merged) located <50 kb around the GWAS variants (downloaded from https://www.ebi.ac.uk/gwas/) was compared with that estimated for a set of randomly sampled common loci (shared with T2T-CHM13) with matched size distribution. The P-values were obtained based on 100 permutations, and enrichments reaching at least marginal significance (BH-adjusted P-value < 0.1) of CPC-specific SVs around (**a**) the GWAS loci reported in global populations and (**b**) those reported in East Asians are shown. In particular, significant enrichments (BH-adjusted P-value < 0.05) are indicated with asterisks (*) on the trait identities. Each boxplot represents the median (thick black line), upper and lower quartiles (box), 1.5× interquartile range (whiskers) of the permutation test statistics (grey dots). The observed statistics are indicated with red dots.

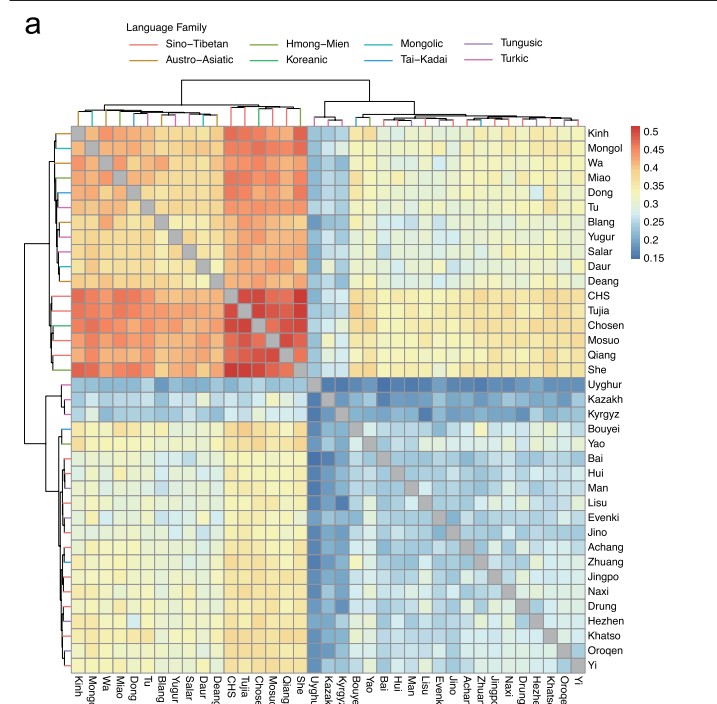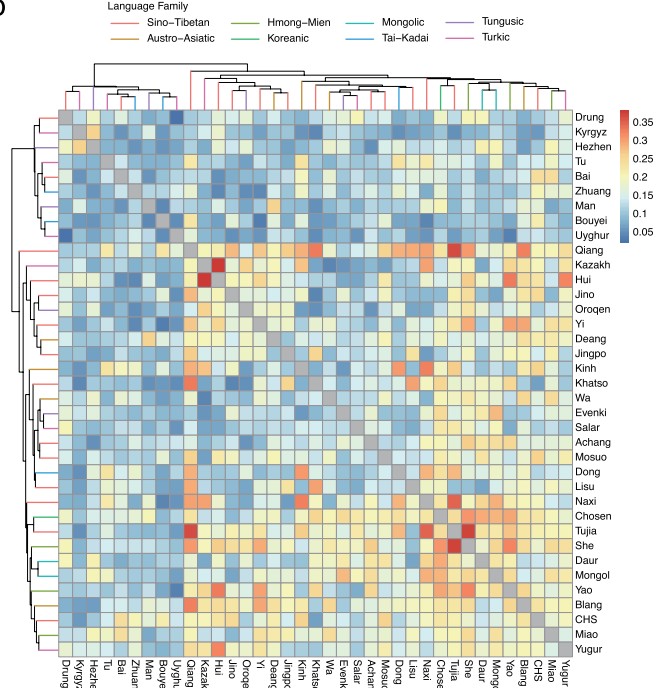

**Extended Data Fig. 10 | Sharing ratio of the archaic ancestry estimated for pairwise CPC populations. a**, Sharing ratio of the Neanderthal-like introgressed sequences. **b**, Sharing ratio of the Denisovan-like introgressed sequences. The heatmaps were generated using the R package pheatmap 1.0.12. Warm colors in the heatmap indicate higher levels of ancestry sharing, while cold colors indicate lower levels of ancestry sharing. Populations are clustered using the complete-linkage method, and the branches are colored according to language families.

# Reporting Summary

## Statistics

For all statistical analyses, confirm that the following items are present in the figure legend, table legend, main text, or Methods section.

| n/a | Confirmed | |
|---|---|---|
| ☐ | ☒ | The exact sample size (*n*) for each experimental group/condition, given as a discrete number and unit of measurement |
| ☐ | ☒ | A statement on whether measurements were taken from distinct samples or whether the same sample was measured repeatedly |
| ☐ | ☒ | The statistical test(s) used AND whether they are one- or two-sided<br>*Only common tests should be described solely by name; describe more complex techniques in the Methods section.* |
| ☐ | ☒ | A description of all covariates tested |
| ☐ | ☒ | A description of any assumptions or corrections, such as tests of normality and adjustment for multiple comparisons |
| ☐ | ☒ | A full description of the statistical parameters including central tendency (e.g. means) or other basic estimates (e.g. regression coefficient) AND variation (e.g. standard deviation) or associated estimates of uncertainty (e.g. confidence intervals) |
| ☐ | ☒ | For null hypothesis testing, the test statistic (e.g. *F*, *t*, *r*) with confidence intervals, effect sizes, degrees of freedom and *P* value noted<br>*Give P values as exact values whenever suitable.* |
| ☒ | ☐ | For Bayesian analysis, information on the choice of priors and Markov chain Monte Carlo settings |
| ☒ | ☐ | For hierarchical and complex designs, identification of the appropriate level for tests and full reporting of outcomes |
| ☐ | ☒ | Estimates of effect sizes (e.g. Cohen's *d*, Pearson's *r*), indicating how they were calculated |

*Our web collection on statistics for biologists contains articles on many of the points above.*

## Software and code

Policy information about availability of computer code

| Data collection | No |
|---|---|
| Data analysis | The code to reproduce the pangenome from this work can be found on GitHub, https://github.com/Shuhua-Group/Chinese-Pangenome-Consortium-Phase-I. Relevant commands used in other analysis can be found in the methods or supplementary information.<br><br>Other software:<br>SMRTLink v9.0<br>ccs-v6.3.0<br>hifiasm v0.16.1<br>QUAST v5.2.0<br>inspector v1.2<br>RepeatMasker1 v4.1.2-p1<br>RMBLAST v2.10.0<br>Dfam2 v3.3<br>dna-brnn3 v0.1<br>minimap2 v2.24<br>PAV v1.2<br>SV-pop v3.0<br>primatR v0.1.0<br>liftoff v1.6.3 |

GATK v4.1.7.0
bwa v0.7.17
Minigraph v0.19
dna-brnn v0.1
Cactus v2.1.1
vg v1.42
hal2vg v1.0.17
GFAffix v0.1.3
RIdeogram v0.2.2
gfabase v0.6.0
bandage v0.9
GraphAligner v1.0.16
gggenes v0.3.1
SNPRelate v1.6.4
ADMIXTURE v1.3.0
samtools v1.15.1
clusterProfiler v3.10.1
ArchaicSeekerV2.0
bcftools v1.14
SVision v1.3.7
pheatmap 1.0.12

For manuscripts utilizing custom algorithms or software that are central to the research but not yet described in published literature, software must be made available to editors and reviewers. We strongly encourage code deposition in a community repository (e.g. GitHub). See the Nature Portfolio guidelines for submitting code & software for further information.

# Data

Policy information about availability of data

All manuscripts must include a data availability statement. This statement should provide the following information, where applicable:
- Accession codes, unique identifiers, or web links for publicly available datasets
- A description of any restrictions on data availability
- For clinical datasets or third party data, please ensure that the statement adheres to our policy

The release of the CPC Phase I data by this work is permitted by The Ministry of Science and Technology of the People's Republic of China (permission no. 2022BAT2392) at the National Genomics Data Center (https://ngdc.cncb.ac.cn) under the BioProject PRJCA011422. The Pangenome References built based on the CPC core samples and combined with the HPRC samples are freely available at both the CPC website https://pog.fudan.edu.cn/cpc/#/data and GitHub (https://github.com/Shuhua-Group/Chinese-Pangenome-Consortium-Phase-I).

Other dataset:
GRCh38 reference (https://ftp.ncbi.nlm.nih.gov/genomes/all/GCA/000/001/405/GCA_000001405.15_GRCh38/seqs_for_alignment_pipelines.ucsc_ids/GCA_000001405.15_GRCh38_no_alt_plus_hs38d1_analysis_set.fna.gz),
T2T-CHM13  reference (https://s3-us-west-2.amazonaws.com/human-pangenomics/T2T/CHM13/assemblies/GCA_009914755.4/chm13v2.0.fa.gz),
HPRC data (https://github.com/human-pangenomics/hpgp-data),
GENCODE v38 (https://www.gencodegenes.org/human/release_38.html),
GIAB 3.0 (https://ftp-trace.ncbi.nlm.nih.gov/ReferenceSamples/giab/release/genome-stratifications/v3.0/GRCh38/),
gnomAD 1.6 (https://gnomad.broadinstitute.org/downloads),
1000 GP phase3 (https://www.ebi.ac.uk/ena/browser/view/PRJEB31736),
Denisovan (https://www.eva.mpg.de/genetics/genome-projects/denisova),
AltaiNean (https://www.eva.mpg.de/genetics/genome-projects/neandertal),
KEGG (https://www.genome.jp/kegg/),
GO (http://geneontology.org/).

# Human research participants

Policy information about studies involving human research participants and Sex and Gender in Research.

| | |
|---|---|
| Reporting on sex and gender | Gender of each sample is reported in the manuscript. we strived to maintain the balance of gender ratio in the sample selection of each population. |
| Population characteristics | The full set of CPC Phase I assemblies include 68 samples representing 36 Chinese minority ethnic groups and 8 linguistic groups. Apart from ethnicity and gender, no other Population characteristics are considered. |
| Recruitment | All 731 randomly collected samples were performed for the whole genome next generation sequencing. To select representative samples for third-generation sequencing, we applied a procedure to quantitatively evaluate the genetic diversity coverage based on PCA results. We selected individuals using a statistic Dd to measure the representation of population samples, it turned out that the selected samples are located close to the center of the cluster of each population on the plot of top two PCs. However, this sample selection method and small sequencing scale may underestimate the true genetic diversity of the population. |
| Ethics oversight | Informed consent was obtained from all individual participants included in the study. The personal identifiers of all samples, if any existed, were stripped off before sequencing and analysis. All procedures were in accordance with the ethical standards |

of the Responsible Committee on Human Experimentation and the 1964 Helsinki Declaration, its later amendments (2000) or comparable ethical standards. The research content and procedures performed in studies involving human participants were approved by the Biomedical Research Ethics Committee of Shanghai Institutes for Biological Sciences (No. ER-SIBS-261408), the Biomedical Research Ethics Committee of Kunming Institute of Zoology, Chinese Academy of Sciences (No. SMKX-20180715-154), the Biomedical Research Ethics Committee of the First Affiliated Hospital of Xi'an Jiaotong University (No. XJTU1AF2021LSK-051).

Note that full information on the approval of the study protocol must also be provided in the manuscript.

# Field-specific reporting

Please select the one below that is the best fit for your research. If you are not sure, read the appropriate sections before making your selection.

☒ Life sciences ☐ Behavioural & social sciences ☐ Ecological, evolutionary & environmental sciences

For a reference copy of the document with all sections, see nature.com/documents/nr-reporting-summary-flat.pdf

# Life sciences study design

All studies must disclose on these points even when the disclosure is negative.

| | |
|---|---|
| Sample size | We followed the HPRC to determine the sample size and applied a procedure similar to that of HPRC to select the representative samples of a subpopulation. For the Phase I of CPC, we selected 68 samples from 731 individuals with genomes deep-sequenced using next-generation sequencing. |
| Data exclusions | We removed 3 samples and 7 samples with relatively low assembly quality from primary assemblies and diploid assemblies, respectively. |
| Replication | Repetition is mostly at the individual level. For example, 1. 1-3 samples were selected from each population to perform third-generation sequencing, 2. 10 samples were selected to evaluate the power of short sequence mapping. |
| Randomization | All the samples are randomly collected from natural populations. In the study, only the population and gender of the sample were considered, and no other phenotypic information was involved. |
| Blinding | At different stages such as sampling, sequencing, and analysis, samples have different ID codes. All the analyses have been applied to all the population groups equally. |

# Reporting for specific materials, systems and methods

We require information from authors about some types of materials, experimental systems and methods used in many studies. Here, indicate whether each material, system or method listed is relevant to your study. If you are not sure if a list item applies to your research, read the appropriate section before selecting a response.

## Materials & experimental systems

| n/a | Involved in the study |
|---|---|
| ☒ | ☐ Antibodies |
| ☐ | ☒ Eukaryotic cell lines |
| ☒ | ☐ Palaeontology and archaeology |
| ☒ | ☐ Animals and other organisms |
| ☒ | ☐ Clinical data |
| ☒ | ☐ Dual use research of concern |

## Methods

| n/a | Involved in the study |
|---|---|
| ☒ | ☐ ChIP-seq |
| ☒ | ☐ Flow cytometry |
| ☒ | ☐ MRI-based neuroimaging |

## Eukaryotic cell lines

Policy information about cell lines and Sex and Gender in Research

| | |
|---|---|
| Cell line source(s) | 47 samples are stored in the Immortalize Cell Bank of Chinese Ethnics Groups hosted in the Institute of Medical Biology, CAMS.<br><br>Sample   Sex<br>--------------------<br>HIFI032682D M<br>HIFI032585D M<br>HIFI032069D F<br>HIFI032373D M |

HIFI032487D F
HIFI032018D M
HIFI032306D F
HIFI032668D M
HIFI032698D F
HIFI032292D F
HIFI032462D M
HIFI032473D M
HIFI032510D F
HIFI032450D M
HIFI032706D M
HIFI032007D F
HIFI032513D M
HIFI032685D M
HIFI032440D M
HIFI032429D M
HIFI032731D F
HIFI032289D M
HIFI032167D M
HIFI032164D M
HIFI032607D F
HIFI032604D F
HIFI032161D F
HIFI032422D M
HIFI032501D M
HIFI032693D M
HIFI032335D U*
HIFI032420D M
HIFI032567D F
HIFI032662D M
HIFI032453D M
HIFI032586D F
HIFI032302D F
HIFI032591D M
HIFI032349D M
HIFI032454D F
HIFI032097D F
HIFI032103D M
HIFI032529D F
HIFI032711D M
HIFI032182D M
HIFI032566D M
HIFI032692D F

* Uncertain due to inconsistency of self reported sex and that based on genetic data.

| | |
|---|---|
| Authentication | The cell line authentication testing was performed by using comparative analysis of genome sequencing data of the cell line and that of blood sample with the same sample ID. |
| Mycoplasma contamination | All cell lines were tested negative for mycoplasma contamination by polymerase chain reaction-based method and culture assay. |
| Commonly misidentified lines (See ICLAC register) | NA. |

