## [Peer Review File · Nature]

Manuscript Title: A Pangenome Reference of 36 Chinese populations

Reviewer Comments & Author Rebuttals

Reviewer Reports on the Initial Version:

Referee #1 (Remarks to the Author):

Gao et al. reported 116 high-quality and haplotype phased de novo assemblies based on 58 core samples representing 36 minority Chinese ethnic groups. The data demonstrate a remarkable increase in discovering novel or missing sequences when individuals are included from underrepresented minority ethnic groups.

In general, it's a very nice paper that publishes abundant new data generated by state-of-art High-Fidelity sequencing technologies. The statistical methods are generally valid and correctly applied. The study adheres to ethical standards, including ethics committee approval and consent procedure. Their data release is permitted by The Ministry of Science and Technology of the People's Republic of China (that's super great!). The reference list covers the relevant literature in a nice way. In addition, The English writing is of sufficient quality. I would like to see a publication of it.

I also have the following points that will need the authors to take into consideration in their revision: (1) The Discussion section is one of the final parts of a research paper, in which an author describes, analyzes, and interprets their findings. They need to explain the significance of those results and tie everything back to the research questions. The paper reports many important novel findings, but the Discussion section focuses on talking about Phase II and Phase III of the project without further discussing the results. I would suggest the authors reorganize their Discussion section to highlight their findings.

(2) The readers will easily get lost in the numerous numbers, parameters and gene names; it would be good if the authors could think of an interesting scientific question to guide the readers. I am particularly interested in the archaic introgression section. Are there any population-specific regions showing the introgression signal? Have the authors found any differences compared East Asians with HPRC Africans (or Native Americans)?

(3) The authors reported a remarkable increase in the discovery of novel sequences, for example, they found a novel duplication event in the BTN gene family region only in an individual from Blang, while a novel deletion only in Wa. It's good to know the project covers a great genetic diversity of the populations, but it's also possible that some observations would be private mutations that are usually found only in a single individual or a small family. I am wondering if the authors could further confirm some of their novel mutations in a larger dataset, for example, in their 731 individuals with genomes deep-sequenced using next-generation sequencing, if the available NGS data have the resolution. In addition, have the authors compared the consistency of their HiFi data and NGS data?

(4) In the "Populations and Samples" part, the authors stated "we selected 68 samples from 731 individuals with genomes deep-sequenced using next-generation sequencing. It turned out that we applied a procedure similar to that of HPRC to select the representative samples of a subpopulation". I would suggest the authors explain their procedure in detail since their populations

sequenced were different from that in HPRC.

(5) I find an outlier (green colour of Sino-Tibetan) in the upper right of the PCA plot in Figure 1. What's that sample? Is this a good representative sample of a population? Could the authors give a detailed PCA plot in the supplementary showing what the grey dots are?

Minor points:

In Figure 1, "27.Mongol-TH (1)" was included in the Sino-Tibetan language family; should it be put in the Mongolic part?

In Figure 1, "32.Han-S (2)", I guess it means Han-Southern, but on the map, 32 was in Henan province, central China. In contrast, "33. Han-C (2)" was in Guangxi, southern China.

Referee #2 (Remarks to the Author):

The contributions of this paper are two-fold. First the assemblies generated are of high quality, with large N50 and few errors, comparable to the HPRC. Second, together the genomes sequences capture a large porportion of the genetic variation present in China.

I find that the part of the paper that deals with generation of the assemblies and the pangenome are well described and represent a considerable amount of work, although the technical novelty is not the key point. I have several suggestions to improve the paper which I believe the authors should be able to address in a major revision.

major issues

1. page 4

The need to improve ... is well known. There is an urgency to establish ...

It would be better if you clearly laid out the problems and how the pangenome aims to resolve them. What is the goal of this paper and how do you achieve it?

2. The methods section needs to be cleaned up (issues below like runlocal etc.) so that the appropriate context is always there. Also there are p-values computed without effect and there is no mention in the methods section of how this was done, e.g. the "sequence alignment" part of the results, in fact there is no corresponding chapter in the methods section that clarifies how the evaluation was performed.

3. Data availability and code

The github repositories contain at this point over 20 repositories and none of them are labelled with either the paper or the term "pangenome".

Similarly the data availability statement claims that "The release of the CPC Phase I data by this work is permitted by The Ministry of Science and Technology of the People's Republic of China

(permission no. 2022BAT2142, 2022BAT1947, and 2022BAT1948) at the National Genomics Data Center (<https://ngdc.cncb.ac.cn>)"

At the National Genomics Data Center link there is no clear link to the pangenomes reported on in this paper.

To put it bluntly, the core product of this paper, the pangenome itself, is nowhere to be found.

4. I find that with this amount of material, i.e. diploid assemblies, diverse populations, etc. there should be something that highlights the advantages of this approach. Inevitably this paper will always be compared to the HPRC paper, indeed the authors directly compare the two graphs in the paper, and in that paper one of the important points was how a new pangenome will improve downstream methods e.g. rna-seq and atac-seq.

The only downstream comparison found in this paper is the number of short reads aligned and variants called. However this part of the paper is weak, since it is presented only with p-values and not an effect estimate (number of reads mapping) and we do not see how many more small variants are called and whether the improvement will translate into additional insights.

I am not asking for additional physical experiments, much of this could be done with publicly available data just as the authors used persons of east asian descent from the 1KG consortium. But I feel there is a gap in the paper here that needs to be filled.

Minor issues

page 2

Despite ... has
has -> having

page 3

the two only -> the only two

page 4

assessed the quality of each assembly
How? either name the methods or refer to the methods section

page 4

CCS
first mention in the paper, never defined

page 5

average ~ 24.34Mb

Contigs or reads? Is the number in the parentheses the min and max? Why is there a "~" used?

page 5

We then evaluated there an average of 3,627 assembly errors

We evaluated the assemblies and discovered an average of

How was this evaluation done? reference to a program or methods section.

page 5

we aligned the 116 assemblies ... and gained the small variants and SVs of each sample

What does gained mean here, probably needs a rewrite.

page 8

Sequence alignment

- There is no mention of the number of reads aligned to either graph genome or a comparison to the reference genome.

- For tandem repeat regions what proportion of reads mapped and how was the comparison to "reference insertion length" computed and compared.

- For each of the results of increase only a p-value is reported but not the effect or what the comparison was. This makes it impossible for the reader to evaluate how much of an improvement this is

- For length of inserte fragments .. either insert length ... or standard deviation. Which is closer to the whole genome level. Do you mean both?

page 8

cactus genome aligne

-> aligner

all paths longer than .. that wasn't

that weren't

page 9

scattered over just one haplotype

Which haplotype and how good was the assembly. If this is not driven by noise, which data point in the PCA plot is this. When a third of the non-ref sequences are from one (half) of an individual it is worth diving into why.

page 10

The cumulatite length .. grows much faster

How much faster? Why isn't there a supplement figure showing the comparison

page 10

We identified 6M small variants and 25K SVs which are novel.

- How is novel defined here, with respect to the HPRC graph? How many of the small variants can be found in gnomad?

- What are the percentage signs in parentheses referring to?

page 10

unreported -> novel

page 12

The CPC core assemblies ??? ~

missing verb

page 14

despite regional efforts are crucial

Rewrite

The paragraph on page 14, "China has been ... communities" adds little to the paper, it supports none of the results and does not put the results into context. I would recommend either shortening it or removing it from the discussion.

page 21

this bam file will be used

The BAM file has not been mentioned up to this point. The tense is wrong, it is used.

runlocal.sh

Without reference to a specific code repository this does not make sense. Either say that you ran this with snakemake or give the parameters to PAV (which are not shown)

szro(0.8,,4)

What is the meaning of these parameters? Also remove runlocal.sh

page 26

Figure 1, legend

T2T-CHM13 and the contigs of ...

The contigs of GRCh38 and T2T-CM13

The X-axis represents the length of the genome unaligned to the GRCh38 ...

Rewrite as the "total length of the genome that could not be aligned to ..."

Figure 2, legend

(c) Venn plot -> Venn diagram

Referee #3 (Remarks to the Author):

Gao et al described the construction and the analysis of a Chinese pangenome, using tools developed by the authors and by the Human Reference Pangenome Consortium (HPRC). The pangenome reveals small and complex variations only observed in Chinese. It sets an example of population-specific pangenome analysis and suggests the importance of sample sizes. Critical observations are as follows:

1) Data availability. The data have been deposited to Chinese National Genomics Data Center (NGDC). However, no download links or accession numbers are given in the manuscript. I can find accession numbers in Reporting Summary, but apparently these data haven't been made public yet. As an important contribution of this manuscript is to provide a resource, the authors should make raw sequences and/or assemblies public if policy allows.

2) Comparison to the HPRC data. My understanding is that the authors constructed a Chinese pangenome graph with minigraph+cactus and then compared the resulting variant calls to the variant calls from HPRC. This is not the best way to compare the two pangenomes because variant representations may differ especially in complex regions. It is likely to overestimate differences between the two pangenomes. The right approach for comparison is to construct a graph jointly from all assemblies and analyze the path differences. This is similar to joint calling when we compare SNP differences across populations.

3) Consistency between variant calls with different methods. The authors called small and structural variants with two approaches: from assembly-to-reference alignment using PAV/SVision (ref-based) vs from minigraph-cactus pangenome graphs (graph-based). They found ~5.8M small variants per sample with ref-based and 4.4M with graph-based. What causes the difference? Which is more correct? Similarly, the authors found 34k structural variations (SVs) per sample with ref-based and 14.5k SVs per haplotype. These numbers don't add up, either. I tend to believe ref-based is an overestimate on both small variants and SVs. If it were me, I would take the graph-based as the main results and use ref-based for validation with an explanation to why ref-based called more.

4) Validating the assemblies. The Chinese pangenome is assembled without parents and has lowered coverage. The overall assembly quality is probably not as good as the HPRC assembly. It is important to validate these assemblies. The authors mainly used QAST for measuring large-scale misassemblies. This is not right. QAST would classify large SVs as misassemblies and thus greatly overestimate misassemblies. It should be avoided in the context of this manuscript. Please use more proper tools like Inspector, Asset (doi:10.1101/2022.05.10.491304) or Flagger (github.com/mobinasri/flagger/).

Minor comments

5) The abstract mentions that Chinese assemblies cover 96.54% of GRCh38 and 93.59% of CHM13. These numbers are mostly affected by incapability of mappers around centromeres. They are an underestimate and misleading. I recommend to remove the numbers from the abstract. Similarly, the authors should be careful of the discussions on the alignment coverage in the main text.

6) The authors should explicitly explain how "novel sequences" are defined. Depending on the criteria, "novel sequences" could range from a few Mb to >100 Mb in length. Personally, I tend to avoid this confusing terminology and just say how many sequences are unaligned under what criteria.

7) The minigraph-cactus pipeline has been recently simplified. Please check out its recent preprint on bioRxiv.

Author Rebuttals to Initial Comments:

January 19, 2023

RE: 2022-08-13204A.R1

Dear Reviewers,

We are grateful to you for your valuable time in evaluating our paper. We would like to thank you for your constructive comments and suggestions which offered a great opportunity for a significant improvement of our manuscript.

We have made corrections which we hope meet with approval. We prepared a separate document and provide a point-to-point response to the comments and also an extra copy of the manuscript with all the changes highlighted.

Thank you very much again for your time.

Sincerely yours

Dr. Shuhua Xu, on behalf of the CPC members
Distinguished Professor & Principal Investigator
Human Population Omics Group
School of Life Sciences & Human Phenome Institute, Fudan University

Point-to-point response to Referees' comments

Authors' Response to Referee #1

Referee #1 (Remarks to the Author):

I also have the following points that will need the authors to take into consideration in their revision:

Comment (1) The Discussion section is one of the final parts of a research paper, in which an author describes, analyzes, and interprets their findings. They need to explain the significance of those results and tie everything back to the research questions. The paper reports many important novel findings, but the Discussion section focuses on talking about Phase II and Phase III of the project without further discussing the results. I would suggest the authors reorganize their Discussion section to highlight their findings.

Response: We thank the reviewer for this constructive comment. Following the suggestion, we have carefully rewritten the Discussion section with more detailed interpretations of the main findings in this work. In particular, we highlighted the novel discoveries of gene variations with the CPC pangenome reference, which further facilitate the evolutionary and medical studies in populations of Asian ancestry.

Comment (2) The readers will easily get lost in the numerous numbers, parameters, and gene names; it would be good if the authors could think of an interesting scientific question to guide the readers. I am particularly interested in the archaic introgression section. Are there any population-specific regions showing the introgression signal? Have the authors found any differences compared East Asians with HPRC Africans (or Native Americans)?

Response: We are grateful for the comments of the reviewer, we reorganized the Results in a question-oriented approach. Regarding the archaic introgression as the reviewer especially pointed out, we have done some extra analyses following the suggestions of the reviewer.

First, compared with non-African population samples, very trivial archaic introgression segments (AISs) have been identified in African samples in the HPRC dataset (Figure R1).

Second, when analyzing the HPRC samples collected from the Americas, we found the proportion of the Altai Neanderthal-like AISs was higher in the Americas than in the CPC East Asians on both individual and population levels (given comparable sample size), while the Denisovan-like AIS proportion is higher in the latter, indicating greater AIS diversity inherited from the Denisovan in the East Asian genomes than in the American genomes (Figure R1).

Third, we found that archaic introgression in East Asians was largely underrepresented in HPRC Asian samples. Each population in the CPC assembly on average added 15.45 Mb AISs to the archaic sequence pool of the present-day East Asians (Figure R2), with every single sample contributing ~9.5 Mb sequences of archaic ancestry. In particular, the Turkic-speaking populations from northwestern China (e.g., Uyghur, Kazakh, and Kyrgyz) showed the least Altai Neanderthal-like AIS sharing with other East Asian populations, while some southern Chinese linguistic groups (e.g., Tai-Kadai and Austro-Asiatic) added to the Denisovan-like AIS diversity on the largest level (Figure R3).

Finally, we further explored the potential biological functions of the CPC-specific archaic introgression regions. As demonstrated in the original version of the manuscript, we found the CPC-specific AISs (absent in the HPRC) are enriched in some cell metabolic processes and are possibly associated with multiple diseases. Following the reviewer's suggestion, we further looked into the novel AIS detected in each East Asian population in CPC. Outstanding examples and detailed descriptions can be found in the revised manuscript.

Figure R1. The total length of the archaic introgression segments detected in CPC compared with that in HPRC on the levels of (A) individuals and (B) populations.

Figure R2. Accumulative length distribution of the archaic introgression segments in the East Asian populations covered by the CPC.

Figure R3. Sharing ratio estimated for pairwise CPC populations based on (A) the Neanderthal-like sequences and (B) the Denisovan-like sequences, respectively.

Comment (3) The authors reported a remarkable increase in the discovery of novel sequences, for example, they found a novel duplication event in the BTN gene family region only in an individual from Blang, while a novel deletion only in Wa. It's good to know the project covers a great genetic diversity of the populations, but it's also possible that some observations would be private mutations that are usually found only in a single individual or a small family. I am wondering if the authors could further confirm some of their novel mutations in a larger dataset, for example, in their 731 individuals with genomes deep-sequenced using next-generation sequencing, if the available NGS data have the resolution. In addition, have the authors compared the consistency of their HiFi data and NGS data?

Response: We thank the reviewer for the suggestions and understand the concerns. Due to the limitations of short reads (150 bp) sequencing technology and existing analysis methods, it is difficult to accurately call or genotype kilobase-scale SVs, especially in complex repetitive regions from the NGS data. However, we have applied an integrated SV-calling workflow, including Breakdancer, Breakseq2, CNVnator, Lumpy, and Manta, on the NGS data of 731 samples, and found nine samples (including the one reported in the HIFI data) with large deletions at the BTN-genes locus, although the boundaries of these deletions were variable and fell within the lower confidence level defined in the calling method. As a result, we have failed to call large duplications in this locus. Our additional analyses have shown the poor performance of existing analysis methods to determine large SVs at the complex regions from the current NGS data. To verify the consistency of the NGS data with HIFI data in a wider range of dimensions, we turned to a widely reported Asia-dominant SV: a 20-kbp deletion (SEA deletion) in the alpha globin gene cluster located at chromosome 16p13.3

(now updated as a new example in our manuscript). We reported one case of the heterozygous SEA deletion from the HiFi data and identified largely overlapping deletion from 20 NGS samples with different boundaries and lower calling confidence. Furthermore, we checked the previously reported SNP array data that were related to the SEA deletion, and found that all 21 samples carried the SEA deletion alleles. However, we failed to call the 10-kb duplication near the SEA deletion from NGS data correctly, similar to the duplication of the *BTN* genes. The results suggested that our identification of complex SVs from HiFi data has high confidence, while calling these variants from NGS data is less accurate, especially for calling duplications from repeat regions, which is due to the nature of the NGS data itself.

We have also attempted to genotype large SVs from NGS data with Pangenie, using our pangenome as the reference panel. However, the results showed low consistency between NGS and HIFI data for both the deletions and duplications mentioned above. The poor performance of the genotyping methods on large SVs is consistent with the results mentioned in the HPRC paper.

We plan to include the analysis and release of NGS data mainly in future work of CPC (e.g. phase2), and will further investigate how to genotype complex variants from NGS data with their corresponding HiFi data as a supervised reference panel.

Comment (4) In the “Populations and Samples” part, the authors stated “we selected 68 samples from 731 individuals with genomes deep-sequenced using next-generation sequencing. It turned out that we applied a procedure similar to that of HPRC to select the representative samples of a subpopulation”. I would suggest the authors explain their procedure in detail since their populations sequenced were different from that in HPRC.

Response: Following the suggestion, we have revised the description of this part.

Comment (5) I find an outlier (green colour of Sino-Tibetan) in the upper right of the PCA plot in Figure 1. What’s that sample? Is this a good representative sample of a population? Could the authors give a detailed PCA plot in the supplementary showing what the grey dots are?

Response: We thank the reviewer for this comment. The responses to this comment are as follows:

1) What's that sample?

The Sino-Tibetan sample in green color in the upper right of the PCA plot is the Drung population.

2) Is this a good representative sample of a population?

As shown in the results of PCA and ADMIXTURE, the Drung population is probably an isolated population with strong genetic drift and thus becomes an outlier cluster in the upper right of the PCA plot. The green one is the representative sample of the Drung population in the Drung cluster. This sample was selected following the procedure of population selection described in the "Populations and Samples" part of the methods.

3) Could the authors give a detailed PCA plot in the supplementary showing what the grey dots are?

Following the reviewer's suggestion, we have added a detailed PCA plot in the SI to show the background samples of Figure 1A.

Minor points:

Comment (6) In Figure 1, "27. Mongol-TH (1)" was included in the Sino-Tibetan language family; should it be put in the Mongolic part?

Response: The "Mongol-TH" in our study is the Mongol mainly living in Tonghai County in the Yunnan Province of southwestern China. They are the descendants of Mongol population but now have their own culture and customs which were largely influenced by the surrounding populations. The Mongol population in Tonghai is also known as Khatso, and they mainly speak the Katso language which belongs to the Sino-Tibetan language family. To avoid the ambiguity of this population name, we have changed the "Mongol-TH" to "Khatso" in our revised Figure 1 and other related figures and tables.

Comment (7) In Figure 1, "32.Han-S (2)", I guess it means Han-Southern, but on the map, 32 was in Henan province, central China. In contrast, "33. Han-C (2)" was in Guangxi, southern China.

Response: We apologize for this mislabeling, we have made corrections in Figure 1A.

Authors' Response to Referee #2

Referee #2 (Remarks to the Author):

major issues

Comment 1. page 4

The need to improve ... is well known. There is an urgency to establish ... It would be better if you laid out the problems and how the pangenome aims to resolve them. What is the goal of this paper and how do you achieve it?

Response: We thank the reviewer for the comments. Following the suggestion, we have expanded the content and clarified the goal of this paper and the approach to achieve it.

Comment 2. The methods section needs to be cleaned up (issues below like runlocal etc.) so that the appropriate context is always there. Also, there are p-values computed without effect and there is no mention in the methods section of how this was done, e.g. the "sequence alignment" part of the results, in fact, there is no corresponding chapter in the methods section that clarifies how the evaluation was performed.

Response: Thank you for your suggestion. We have cleaned up some code that may confuse. Following the suggestion, we have rewritten the “sequence alignment” part and showed the analysis details in the revised manuscript.

Comment 3. Data availability and code

The GitHub repositories contain at this point over 20 repositories and none of them are labeled with either the paper or the term "pangenome".

Similarly, the data availability statement claims that "The release of the CPC Phase I data by this work is permitted by The Ministry of Science and Technology of the People's Republic of China (permission no. 2022BAT2142, 2022BAT1947, and 2022BAT1948) at the National Genomics Data Center (<https://ngdc.cnbc.ac.cn>)"

At the National Genomics Data Center link, there is no clear link to the pangenomes reported on in this paper.

To put it bluntly, the core product of this paper, the pangenome itself, is nowhere to be found.

Response: Thank you for pointing out these problems. We have tried our best to improve our presentation and provided the necessary information.

First, we made the label more explicit and updated the GitHub (<https://github.com/Shuhua-Group/Chinese-Pangenome-Consortium-Phase-I>) and provided key code, file download, and other support information.

Second, we also provided the BioProject accession numbers in the revised manuscript.

Finally, we have provided the download link of the graph genome on the official website of the Chinese Pangenome Consortium (<https://pog.fudan.edu.cn/cpc/#/data>). Because of the limitation of the file size on the GitHub website, we are not able to upload the Pangenome itself directly to GitHub. However, we provided a link on the GitHub website so that the users can simply jump from the GitHub page (<https://github.com/Shuhua-Group/Chinese-Pangenome-Consortium-Phase-I>).

Comment 4. I find that with this amount of material, i.e. diploid assemblies, diverse populations, etc. there should be something that highlights the advantages of this approach. Inevitably this paper will always be compared to the HPRC paper, indeed the authors directly compare the two graphs in the paper, and in that paper one of the important points was how a new pangenome will improve downstream methods e.g. rna-seq and atac-seq.

The only downstream comparison found in this paper is the number of short reads aligned and variants called. However, this part of the paper is weak, since it is presented only with p-values and not an effect estimate (number of reads mapping) and we do not see how many more small variants are called and whether the improvement will translate into additional insights.

I am not asking for additional physical experiments, much of this could be done with publicly available data just as the authors used persons of east Asian descent

from the 1KG consortium. But I feel there is a gap in the paper here that needs to be filled.

Response: Thank you for pointing out this problem. Following your suggestions, we provided more detailed alignment statistics in the revised version. We also included in the revised version of the manuscript the advantages of using population-specific graph reference in short reads alignment and archaic sequence identification. Moreover, we have improved the analysis and statistics in this part, and discussed some problems with the alignment software and the variants calling in the analysis of the short read. Thank you again for your consideration, we have tried our best and have done pretty much additional analysis with the available 1KG data, as the reviewer recommended, to fill the gap in comparing the applications of the two Pan-genomes.

Minor issues

Comment 5. page 2

Despite ... has
has -> having

Response: Thank you for pointing out this typo, it has been corrected.

Comment 6. page 3

the two only -> the only two

Response: Thank you for pointing out this typo, it has been corrected.

Comment 7. page 4

assessed the quality of each assembly

How? either name the methods or refer to the methods section

Response: We have added the reference to the methods section (see Methods, 'Genome assembly and quality control') at the end of this description.

Comment 8. page 4

Comment 9. CCS

first mention in the paper, never defined

Response: We have added the full name "circular consensus sequencing" in the first mention of CCS.

Comment 10. page 5

average ~ 24.34Mb

Contigs or reads? Is the number in the parentheses the min and max? Why is there a "~" used?

Response: The 24.34Mb here is the length of novel contigs absent from the commonly used reference sequence. The two numbers in the parentheses are minimum and maximum. Since the number was presented using scientific notation, we have used "~"

here to indicate this number is an approximate number rather than an exact number.

Comment 11. page 5

We then evaluated there an average of 3,627 assembly errors

We evaluated the assemblies and discovered an average of ...

How was this evaluation done? reference to a program or methods section.

Response: The evaluation was performed with QUAST and Inspector. We have referred to the Methods sections with detailed information, i.e., "Assembly polishing and assessment" section of "Methods" .

Comment 12. page 5

we aligned the 116 assemblies ... and gained the small variants and SVs of each sample

What does gained mean here, probably needs a rewrite.

Response: Following the suggestion, we have changed our description of this sentence as "we aligned the 116 assemblies to the T2T-CHM13 and used PAV to call the small variants and SVs of two assemblies of each sample" .

Comment 13. page 8

Sequence alignment

– There is no mention of the number of reads aligned to either graph genome or a comparison to the reference genome.

– For tandem repeat regions what proportion of reads mapped and how was the comparison to "reference insertion length" computed and compared.

– For each of the results of increase only a p-value is reported but not the effect or what the comparison was. This makes it impossible for the reader to evaluate how much of an improvement this is

– For length of inserte fragments .. either insert length ... or standard deviation. Which is closer to the whole genome level. Do you mean both?

Response: Thank you for pointing out this problem. We have redesigned this analysis and presented more detailed and insightful results.

Comment 14. page 8

cactus genome aligne

-> aligner

Response: We thank the reviewer for pointing it out. We have corrected the typo in the revised manuscript.

Comment 15. all paths longer than .. that wasn't that weren't

Response: We thank the reviewer for pointing it out. We have corrected the typo in the revised manuscript.

Comment 16. page 9

scattered over just one haplotype

Which haplotype and how good was the assembly. If this is not driven by noise, which data point in the PCA plot is this. When a third of the non-ref sequences are from one (half) of an individual it is worth diving into why.

Response: Thank you for pointing out this ambiguity. We originally intended to describe the number of singletons. We have revised the part in the main text.

Comment 17. page 10

The cumulative length .. grows much faster

How much faster? Why isn't there a supplement figure showing the comparison

Response: Thank you for the suggestion. We have added a supplement figure showing the comparison.

Comment 18. page 10

We identified 6M small variants and 25K SVs which are novel.

– How is novel defined here, with respect to the HPRC graph? How many of the small variants can be found in gnomad?

– What are the percentage signs in parentheses referring to?

Response: We apologize for the unclear description. In the initial manuscript, the “novel variants” referred to the variants that can only be found in the CPC pangenome graph, with respect to variants that can only be found in the HPRC graph, and the percentage signs in parentheses refer to the number of CPC novel variants as the proportion of all that type of variants in CPC pangenome graph. Following the suggestion of the reviewers, we have adopted a more accurate and direct approach to comparing the CPC with the HPRC graph in the updated manuscript by creating a joint graph that includes all assemblies. We now define the “CPC-specific variants” as those found only in the 116 assemblies of CPC (the right half of Fig.3C Venn diagram), relative to the HPRC-specific and common variants (the left half and overlapped part of Fig3C Venn diagram). The percentage signs in parentheses refer to the number of CPC-specific variants as the proportion of all that types of variants in the merged pangenome graph.

Following the suggestion, we have also annotated the CPC-specific small variants with gnomad v0.1.8, and found that about 39% of the CPC-specific small variants could not be found in gnomad. Compared to the HPRC-specific variants, CPC-specific variants that are not observed in gnomad are more enriched in the “easy” regions of GRCh38 reference genome defined in GIAB 3.0 as a guarantee of high confidence, which suggested that the CPC-specific variants would be a powerful addition to the diversity of current human genetic data resource.

Table R1 | The number of variants that can be found in the gnomad v0.1.8 in different regions defined in GIAB 3.0

	CPC_specific	HPRC_specific	common
GIAB_difficult not_gnomad	470,575	695,935	3,652,516
GIAB_difficult gnomad	747,025	2,445,856	3,963,791

GIAB_easy not_gnomad	934,874	622,404	171,494
GIAB_easy gnomad	2,367,509	8,213,124	6,247,382

Comment 19. page 10

unreported -> novel

Response: Thank you for your suggestion. We have made corrections accordingly.

Comment 20. page 12

The CPC core assemblies ??? ~

missing verb

Response: Thank you for pointing out this typo, which has been corrected.

Comment 21. page 14

despite regional efforts are crucial

Rewrite

The paragraph on page 14, "China has been ... communities" adds little to the paper, it supports none of the results and does not put the results into context. I would recommend either shortening it or removing it from the discussion.

Response: Thank you for your suggestion. We have rewritten the Discussion and have removed this particular last paragraph from the revised manuscript.

Comment 22. page 21

this bam file will be used

The BAM file has not been mentioned up to this point. The tense is wrong, it is used.

Response: Thank you for your suggestion. We have corrected the description accordingly.

Comment 23. runlocal.sh

Without reference to a specific code repository this does not make sense. Either say that you ran this with snakemake or give the parameters to PAV (which are not shown)

Response: We apologize for the ambiguity of the method description of PAV. We have rewritten the "Variant identification from phased assembly" in the method section and added the related parameter files of PAV analysis into the code repository of our study.

Comment 24. szro(0.8,,4)

What is the meaning of these parameters? Also, remove runlocal.sh

Response: We are sorry about the ambiguity of the parameters used in SV-pop analysis. In our study, we used SV-pop to merge SVs among the CPC sample. As we described in the old version of the manuscript, the parameters of SV-pop is nr::szro(0.8,,4):match(0.8). The meanings of these parameter are as follows:

- 1) The "nr" means Nonredundant Merging, a merge system built into SV-Pop based on an order of input sources.
- 2) The "szro" means Size Reciprocal Overlap, it is used to identify insertion with size match

within some percentage by size, and it required three parameters. The first parameter is the size overlap rate between insertions. The second parameter is the null value of our SV-pop analysis, it is the maximum distance between insertions. The third parameter is the maximum value of the distance between insertions / the insertion length. Thus, the parameters (0.8,4) means when the distance between two insertions is less than 4 times the insertion length, insertions with at least 80% overlap will be merged.

3) The “match” parameter is used to identify the identity between two insertions. The parameter (0.8) means insertions with an identity higher than 80% will be merged.

4) Following the suggestion, we have removed “runlocal.sh”, rewritten the Method description of PAV, and explained the meaning of these parameters in method section.

Reference: <https://github.com/EichlerLab/svpop/blob/master/MERGE.md>

Comment 25. page 26

Figure 1, legend

T2T-CHM13 and the contigs of ...

The contigs of GRCh38 and T2T-CM13

Response: Thank you for your suggestion. We have revised this sentence in the legend of Figure 1.

Comment 26. The X-axis represents the length of the genome unaligned to the GRCh38 ...

Rewrite as the "total length of the genome that could not be aligned to ..."

Response: Thank you for your suggestion. We have revised this sentence in the legend of Figure 1.

Comment 27. Figure 2, legend

(c) Venn plot -> Venn diagram

Response: Thank you for your suggestion. We have revised accordingly in the legend of Figure 2. We thank the reviewer so much for carefully reading our manuscript and giving so many detailed suggestions.

Authors' Response to Referee #3

Referee #3 (Remarks to the Author):

Comment 1) Data availability. The data have been deposited to Chinese National Genomics Data Center (NGDC). However, no download links or accession numbers are given in the manuscript. I can find accession numbers in Reporting Summary, but apparently these data haven't been made public yet. As an important contribution of this manuscript is to provide a resource, the authors should make raw sequences and/or assemblies public if policy allows.

Response: We apologize for the ambiguous data access information. We have updated the description of the data availability and provided the NGDC BioProject

accession numbers in the revised manuscript. Yes, we confirm that we can make the data public as we have already obtained the official approval.

Comment 2) Comparison to the HPRC data. My understanding is that the authors constructed a Chinese pangenome graph with minigraph+cactus and then compared the resulting variant calls to the variant calls from HPRC. This is not the best way to compare the two pangenomes because variant representations may differ especially in complex regions. It is likely to overestimate differences between the two pangenomes. The right approach for comparison is to construct a graph jointly from all assemblies and analyze the path differences. This is similar to joint calling when we compare SNP differences across populations.

Response: We thank the reviewer for the suggestion. Following the suggestion, we have constructed a new MC pangenome graph including all 116 assemblies in CPC and 94 assemblies in HPRC and compared the difference directly. We have updated the whole “Comparison with HPRC” section as well as the description in the Methods, and the joint pangenome graph is now freely available online.

Comment 3) Consistency between variant calls with different methods. The authors called small and structural variants with two approaches: from assembly-to-reference alignment using PAV/SVision (ref-based) vs from minigraph-cactus pangenome graphs (graph-based). They found ~5.8M small variants per sample with ref-based and 4.4M with graph-based. What causes the difference? Which is more correct? Similarly, the authors found 34k structural variations (SVs) per sample with ref-based and 14.5k SVs per haplotype. These numbers don't add up, either. I tend to believe ref-based is an overestimate on both small variants and SVs. If it were me, I would take the graph-based as the main results and use ref-based for validation with an explanation to why ref-based called more.

Response: We are grateful to the reviewer's comment. We have checked our analysis results and found the differences resulted from the following issues:

i) In ref-based variant-calling analysis, PAV align each of the whole assembly to the T2T-CHM13 reference to discover the variants. However, in graph-based analysis, following the minigraph-cactus pipeline, we used dna-brnn to identify the highly repetitive sequences of each assembly and masked them before the construction of the CPC pangenome graph. Thus, one of the differences between the ref-based and graph-based methods was caused by the ref-based variants in the highly repetitive sequences. We counted that an average of ~0.65M small variants and ~8.26K SVs per CPC sample are in these regions (Figure R5). In the set of variants from ref-based instead of graph-based, the variants in highly repetitive regions accounted for an average of 38.38% for small variants and 53.97% for SVs per sample (Figure R6).

ii) Except for masking highly repetitive sequences before the construction, we removed parts of the sequences of each assembly in two other steps in the MC pipeline. In the “cactus-graph map-split” step, we assigned contigs to chromosomes, in which parts of the contigs failed to align to any chromosomes

in the former step, so they would be classified as “ambiguous” and excluded in the following analysis. Each assembly lost ~27.8 Mbp on average in this step. In the “cactus-graph map-join”, we clipped out unaligned sequences longer than 100 Kbp to avoid indexing error during generating paths, and each assembly lost ~61.9 Mbp on average in this step.

We applied these quality control steps consistency with the HPRC MC graph to make the results comparable. Although most of the removed sequences were barely aligned to the reference genome as well as to the constructed minigraph, this would result in a relatively smaller number of graph-based variants to a considerable extent since we did not perform the corresponding step in PAV.

iii) The variants of the MC graphs counted by us and by HPRC are deconstructed from the topological graph containing all variants of all assemblies, and here is a problem: the nesting of variants at complex loci. If there is a larger-scale complex structural variation at a locus, such as a kb-scale deletion, then the node where this deletion is located would act as the shortest branching path in the topology graph, and the “deleted” sequences of the reference genome and the corresponding sequences of other samples without deletions will be juxtaposed as “bubbles” at this node without detecting the variants between them, which will result in the underestimation of the total amount of variants at complex loci, especially for small variants. There is some current work attempting to resolve the nested variants. For example, we used the unofficially released module “vcfwave” in vcflib to decompose complex variants, and the results showed an average of ~5.55K new SVs as well as ~0.37M new small variants per sample (Figure R7).

iv) For SVision, we used it to call the complex SVs (CSVs), containing multiple breakpoints, based on HiFi reads alignment against the T2T-CHM13 genome and detected 70 CSVs per samples on average. However, graph-based methods are hardly to detect CSVs. Therefore, we are not able to evaluate the consistency of CSVs between SVision and the graph-based method.

In summary, we took additional quality control steps during the construction of the MC pangenome graph and removed some of the sequences from each assembly, and the variants deconstructed from the topological graph are inherently nested; in contrast, we did not use similar measures in the ref-based method PAV, which resulted in the greater number of ref-based variants than those deconstructed directly from the graph. The variants in our three methods differ in their analytical purpose. PAV called variants at the individual level, without merging and filtering in the population level, as a traditional reference metric for assembly quality assessment. MC graph-based variants were deconstructed from the topological graph, which would be used as the reference for subsequent applications (NGS alignment, RNA-seq mapping, SV genotyping, etc.), so that essential quality controls were applied to ensure the performance of the graph; SVision were applied as a state-of-the-art deep-learning method to call CSVs that were difficult to detect by traditional base-alignment or graph-based methods.

Figure R5 Number of variants of CPC samples by aligning two assemblies of each sample to the T2T-CHM13 reference. The number of variants in highly repetitive regions is marked as a shadow in each bar.

Figure R6 Proportion of variants in the highly repetitive regions of the set of variants is from ref-based instead of graph-based.

Figure R7 Number of CPC graph variants after decomposition of complex variants compared with the number of original CPC graph variants. The number of new variants after the decomposition of complex variants in the CPC graph is marked as a shadow in each bar.

Comment 4) Validating the assemblies. The Chinese pangenome is assembled without parents and has lowered coverage. The overall assembly quality is probably not as good as the HPRC assembly. It is important to validate these assemblies. The authors mainly used QUILT for measuring large-scale misassemblies. This is not right. QUILT would classify large SVs as misassemblies and thus greatly overestimate misassemblies. It should be avoided in the context of this manuscript. Please use more proper tools like Inspector, Asset (doi:10.1101/2022.05.10.491304), or Flagger (github.com/mobinasri/flagger/).

Response: We thank the reviewer for reminding us of these issues. Following the suggestion, we have made the further revisions as below:

1) As we described in our manuscript, the misassemblies of our study were mainly measured by the Inspector. In the old version of the manuscript, the results of the misassembly evaluation based on the QUILT were only shown in the SI to compare the misassemblies before and after assembly polishing. Following the reviewer's suggestion, we have removed the results of misassemblies evaluated by the QUILT in the SI.

2) Figure 1E showed the small-scale misassemblies evaluated by the Inspector. We have marked "small-scale" in the assembly error assessment for this result in our revised main text and figure legend. We have also added the evaluation results of large-scale misassemblies performed by the Inspector to the SI.

Minor comments

Comment 5) The abstract mentions that Chinese assemblies cover 96.54% of GRCh38 and 93.59% of CHM13. These numbers are mostly affected by the incapability of mappers around centromeres. They are an underestimate and misleading. I recommend removing the numbers from the abstract. Similarly, the authors should be careful of the discussions on the alignment coverage in the main text.

Response: Following the suggestion, we have removed the numbers from the Abstract, and we checked carefully the main text and discussion as well.

Comment 6) The authors should explicitly explain how "novel sequences" are defined. Depending on the criteria, "novel sequences" could range from a few Mb to >100 Mb in length. I tend to avoid this confusing terminology and just say how many sequences are unaligned under what criteria.

Response: The novel sequences were defined as insertions larger than 1 kbp with respect to the T2T-CHM13 reference. We have more explicitly defined the "novel sequences" when it was first mentioned in our manuscript.

Comment 7) The minigraph-cactus pipeline has been recently simplified. Please check out its recent preprint on bioRxiv.

Response: We thank the reviewer for reminding us of this update. We have checked the preprint paper and release history of the MC pipeline, and found no major functional updates to the latest version compared to the one we used. Therefore, we have used the version with a simplified procedure as the reviewer mentioned. We have updated the citation in our manuscript.

Reviewer Reports on the First Revision:

Referee #1 (Remarks to the Author):

The authors have addressed all my concerns. I am happy to see the publication of this paper.

Referee #2 (Remarks to the Author):

I found the responses to my comments to be satisfactory for the most part. There are a few things that can be cleared up below. I ask the authors to please indicate the precise changes in the rebuttal as much of the re-reviewing required me to compare diffs to see what the changes were and in some cases text was deleted but not indicated in the rebuttal

> Comment 2. The methods section needs to be cleaned up (issues below like runlocal etc.) so that the appropriate context is always there. Also, there are p-values computed without effect and there is no mention in the methods section of how this was done, e.g. the "sequence alignment" part of the results, in fact, there is no corresponding chapter in the methods section that clarifies how the evaluation was performed.

> Response: Thank you for your suggestion. We have cleaned up some code that may confuse. Following the suggestion, we have rewritten the "sequence alignment" part and showed the analysis details in the revised manuscript.

While much of the methods section has been improved there is still not description of how p-values were computed. Another comment I had (Comment 13), had to do with computation of p-values w.r.t. mapped reads, but throughout the paper there are claims of statistical significance where the methods are missing, e.g. all mentions of enrichment from GO categories, signatures of natural selection, enrichment of SNPs associated with disease, etc. Furthermore the p-values are listed without effect sizes, (except for Tajima's D) which make it hard for the reader to put them in context.

>Comment 3. Data availability and code

The GitHub repositories contain at this point over 20 repositories and none of them are labeled with either the paper or the term "pangenome". Similarly, the data availability statement claims that "The release of the CPC Phase I data by this work is permitted by The Ministry of Science and Technology of the People's Republic of China (permission no. 2022BAT2142, 2022BAT1947, and 2022BAT1948) at the National Genomics Data Center (<https://ngdc.cncb.ac.cn>)" At the National Genomics Data Center link, there is no clear link to the pangenomes reported on in this paper. To put it bluntly, the core product of this paper, the pangenome itself, is nowhere to be found.

>Response: Thank you for pointing out these problems. We have tried our best to improve our presentation and provided the necessary information. First, we made the label more explicit and updated the GitHub (<https://github.com/Shuhua-Group/Chinese-Pangenome-Consortium-Phase-I>) and provided key code, file download, and other support information. Second, we also provided the

BioProject accession numbers in the revised manuscript. Finally, we have provided the download link of the graph genome on the official website of the Chinese Pangenome Consortium (<https://pog.fudan.edu.cn/cpc/#/data>). Because of the limitation of the file size on the GitHub website, we are not able to upload the Pangenome itself directly to GitHub. However, we provided a link on the GitHub website so that the users can simply jump from the GitHub page (<https://github.com/Shuhua-Group/Chinese-Pangenome-Consortium-Phase-I>).

I have checked the data provided, and the pangenome graph is publicly available, the accession numbers list the assemblies, but have not yet been released (scheduled for 2026), I assume they will be released when the paper is published. The code released contains two scripts that can be considered somewhat complete for PAV and SV-pop but release a supplementary document listing the commands used for running main parts of the pipeline in order to construct the pangenome.

On pg 30, the authors state "All of the computer code and scripts necessary to reproduce the results from this work can be found on (url)". This would be interpreted that all the results are reproducible from the code repository, however only the construction of the pangenome is described. The code as organized can be used to reproduce the pangenome by copying and adapting the relevant commands, but this does not meet the strict standards of automatic reproducibility. I would suggest that the authors rephrase this to reflect that the code to reproduce the pangenome is available. This is to the same standard as a list of command lines used in a methods section or supplement (which is perfectly fine, but the text should reflect it)

>Comment 13. page 8

>Sequence alignment

>- There is no mention of the number of reads aligned to either graph genome or a comparison to the reference genome.

>- For tandem repeat regions what proportion of reads mapped and how was the comparison to "reference insertion length" computed and compared.

>- For each of the results of increase only a p-value is reported but not the effect or what the comparison was. This makes it impossible for the reader to evaluate how much of an improvement this is

>- For length of insert fragments .. either insert length ... or standard deviation. Which is closer to the whole genome level. Do you mean both?

>Response: Thank you for pointing out this problem. We have redesigned this analysis and presented more detailed and insightful results.

Judging from the diff it seems that this section was removed from the paper, I cannot find it in the supplement or elsewhere in the paper. In the response can you list how the analysis was redesigned, where it was presented in more detail and what additional insights were found?

>Comment 16. page 9

>scattered over just one haplotype

>Which haplotype and how good was the assembly. If this is not driven by noise, which data point in the PCA plot is this. When a third of the non-ref sequences are from one (half) of an individual it is worth diving into why.

>Response: Thank you for pointing out this ambiguity. We originally intended to describe the number of singletons. We have revised the part in the main text.

This is a revision by deleting text. Even though 60Mb of sequences are singletons is the distribution even across samples or is it driven by outliers?

> Comment 18

About the gnomad variants.

I appreciate that you looked this up and provided the table but I think you missed an opportunity to drive home the main point of your paper. In the summary/abstract you write "The CPC data demonstrate a remarkable increase in discovering novel or missing sequences when individuals are included from underrepresented minority ethnic groups". A devil's advocate might argue that there is less need for a pangenome when short read sequences from a larger group of individuals will capture more small variants and they have already been captured in gnomad. Based on the table you can argue that you add a significant number of variants not previously identified in gnomad and not captured by HPRC, both in the easy and difficult GIAB regions.

Referee #3 (Remarks to the Author):

In this revision, the authors have addressed most of my concerns. In particular, they constructed a joint pangenome with HPRC and made the pangenome graphs public. The comparison to HPRC makes sense now and is quite interesting. However, the authors haven't addressed all my concerns. I still have one major comment and several minor comments.

Major comment

1) Some of the PAV numbers are likely to be wrong. The authors found 4.72 million SNPs per sample. With short reads, we typically call 3.3-3.5 million SNPs for non-Africans. With long reads we should call more, but 4.72 million is still too large. PAV seems to be aggressively making calls in regions it doesn't have enough power. Probably the majority of additional SNP calls are false positives. At the bottom line, the large difference between PAV-based calls (5.8M) and MC-based calls (4.4M) would confuse readers. I suggest the authors not describe PAV-based small variant calls. If the number of SNP calls is not solid, the number of SV calls with PAV is also questionable. The authors may replace Extended Data Fig 3 with MC-based results. It is ok to describe the SVision results.

Minor comments

2) I can see the authors have uploaded the contig sequences but they haven't made them public. I trust the authors to release the data once the paper is published.

3) Could the authors run RepeatMasker on unmapped and novel sequences? Probably the majority are centromeric repeats. Would be good to report the number.

4) The authors are using inspector to evaluate assemblies, as I suggested earlier. It is worth mentioning that all these tools have false positives, reporting correctly assembled regions as errors. The authors may mention this in the main text. Honestly, I am surprised by the high number. If it were me, I would look at the read alignment in IGV to understand what is happening.

5) The authors found many novel SVs that are close to GWAS hits. It would be good to evaluate the significance of these hits with, for example, permutation test.

Author Rebuttals to First Revision:

March 31, 2023

RE: 2022-08-13204C.R3

Dear Reviewers,

We are grateful to you for your valuable time in evaluating our paper. We would like to thank you for your constructive comments and suggestions which offered a great opportunity for a significant improvement of our manuscript.

We have made corrections which we hope meet with approval. We prepared a separate document and provide a point-to-point response to the comments and also an extra copy of the manuscript with all the changes highlighted.

Thank you very much again for your time.

Sincerely yours

Dr. Shuhua Xu, **on behalf of the CPC members**
Distinguished Professor & Principal Investigator
Human Population Omics Group
School of Life Sciences & Human Phenome Institute, Fudan University

Point-to-point response to Referees' comments

Authors' Response to Referee #1

Referee #1 (Remarks to the Author):

Referees' comments:

Referee #1 (Remarks to the Author):

The authors have addressed all my concerns. I am happy to see the publication of this paper.

Response: We thank the reviewer for such a great support.

Authors' Response to Referee #2

Referee #2 (Remarks to the Author):

I found the responses to my comments to be satisfactory for the most part. There are a few things that can be cleared up below. I ask the authors to please indicate the precise changes in the rebuttal as much of the re-reviewing required me to compare diffs to see what the changes were and in some cases text was deleted but not indicated in the rebuttal

Response: We apologize that we did not make it clear when we answered those questions in the previous reply. We try our best to carefully address the concerns of the reviewer.

> Comment 2. The methods section needs to be cleaned up (issues below like runlocal etc.) so that the appropriate context is always there. Also, there are p- values computed without effect and there is no mention in the methods section of how this was done, e.g. the "sequence alignment" part of the results, in fact, there is no corresponding chapter in the methods section that clarifies how the evaluation was performed.

> Response: Thank you for your suggestion. We have cleaned up some code that may confuse. Following the suggestion, we have rewritten the "sequence alignment" part and showed the analysis details in the revised manuscript.

While much of the methods section has been improved there is still not description of how p-values were computed. Another comment I had (Comment 13), had to do with computation of p-values w.r.t. mapped reads, but throughout the paper there are claims of statistical significance where the methods are missing, e.g. all mentions of enrichment from GO categories, signatures of natural selection, enrichment of SNPs associated with disease, etc. Furthermore the p-values are listed without effect sizes, (except for Tajima's D) which make it hard for the reader to put them in context.

Response: We thank the reviewer for pointing out this. We provide details of the statistical tests in the revised manuscript (lines 696-701, 718-719; page 8, lines 898-901, 909-914, page 13; lines 989-990, 997-1001, page 15; lines 1070-1076, page 16; lines 1133-1135, page 17) and the corresponding supplementary information (Table S4, S5, S10, S11, S16, S18-20, S22).

>Comment 3. Data availability and code

The GitHub repositories contain at this point over 20 repositories and none of them are labeled with either the paper or the term "pangenome". Similarly, the data availability statement claims that "The release of the CPC Phase I data by this work is permitted by The Ministry of Science and Technology of the People's Republic of China (permission no. 2022BAT2142, 2022BAT1947, and 2022BAT1948) at the National Genomics Data Center (<https://ngdc.cncb.ac.cn>)" At the National Genomics Data Center

link, there is no clear link to the pangenomes reported on in this paper. To put it bluntly, the core product of this paper, the pangenome itself, is nowhere to be found.

>Response: Thank you for pointing out these problems. We have tried our best to improve our presentation and provided the necessary information. First, we made the label more explicit and updated the GitHub (<https://github.com/Shuhua-Group/Chinese-Pangenome-Consortium-Phase-I>) and provided key code, file download, and other support information. Second, we also provided the BioProject accession numbers in the revised manuscript. Finally, we have provided the download link of the graph genome on the official website of the Chinese Pangenome Consortium (<https://pog.fudan.edu.cn/cpc/#/data>). Because of the limitation of the file size on the GitHub website, we are not able to upload the Pangenome itself directly to GitHub. However, we provided a link on the GitHub website so that the users can simply jump from the GitHub page (<https://github.com/Shuhua-Group/Chinese-Pangenome-Consortium-Phase-I>).

I have checked the data provided, and the pangenome graph is publicly available, the accession numbers list the assemblies, but have not yet been released (scheduled for 2026), I assume they will be released when the paper is published. The code released contains two scripts that can be considered somewhat complete for PAV and SV-pop but release a supplementary document listing the commands used for running main parts of the pipeline in order to construct the pangenome.

On pg 30, the authors state "All of the computer code and scripts necessary to reproduce the results from this work can be found on (url)". This would be interpreted that all the results are reproducible from the code repository, however only the construction of the

pangenome is described. The code as organized can be used to reproduce the pangenome by copying and adapting the relevant commands, but this does not meet the strict standards of automatic reproducibility. I would suggest that the authors rephrase this to reflect that the code to reproduce the pangenome is available. This is to the same standard as a list of command lines used in a methods section or supplement (which is perfectly fine, but the text should reflect it)

Response: Thank you for your suggestion. We have reworded and revised this section. Please see "Data availability" section (lines 1724-1737, page 30).

>Comment 13. page 8

>Sequence alignment

>- There is no mention of the number of reads aligned to either graph genome or a comparison to the reference genome.

>- For tandem repeat regions what proportion of reads mapped and how was the comparison to "reference insertion length" computed and compared.

>- For each of the results of increase only a p-value is reported but not the effect or what the comparison was. This makes it impossible for the reader to evaluate how much of an improvement this is

>- For length of insert fragments .. either insert length ... or standard deviation. Which is closer to the whole genome level. Do you mean both?

>Response: Thank you for pointing out this problem. We have redesigned this analysis and

presented more detailed and insightful results.

Judging from the diff it seems that this section was removed from the paper, I cannot find it in the supplement or elsewhere in the paper. In the response can you list how the analysis was redesigned, where it was presented in more detail and what additional insights were found?

Response: We apologize for not explaining it clearly. We have removed the original section. The new results are in the "Short-read mapping with the CPC reference" section (lines 823-865, page 11). For method descriptions, see the "Evaluation of the alignment quality of short reads using the CPC reference" section (lines 1666-1684, page 29).

In the original results, we subjected the ".gam" file generated by "vg giraffe" to the CHM13 linear reference coordinate, and then evaluated the ".bam". However, in subsequent analysis, we found that the subject step would result in a loss of specific information from the graphical genome. For this reason, we conducted a new assessment at the ".gam" level.

In the new results, we found that the mapping rate is greatly affected by the complexity of the graph. Secondly, using population specific pangenome references can improve the quality of alignment. Finally, we also propose the loss of graph specific information caused by the subject step.

>Comment 16. page 9

>scattered over just one haplotype

>Which haplotype and how good was the assembly. If this is not driven by noise, which data point in the PCA plot is this. When a third of the non-ref sequences are from one (half) of an individual it is worth diving into why.

>Response: Thank you for pointing out this ambiguity. We originally intended to describe the number of singletons. We have revised the part in the main text.

This is a revision by deleting text. Even though 60Mb of sequences are singletons is the distribution even across samples or is it driven by outliers?

Response: We thank the reviewer for asking, indeed, it was also our concern to understand the distribution of the non-ref sequences among individual samples. Following the suggestion of the reviewers, we have calculated length of the non-ref singleton from each assembly. The results show that aggregation did occur. In particular, the maximum 7.04 Mb was observed in a Han Chinese sample from the HPRC. We have checked the data carefully and try to find out what happened. It turned out that the long non-ref sequences are located in complex regions of the genome with a lot of simple repeats. Although it has some considerable local similarity to the reference sequence, overall it cannot be finely parsed when constructing a graph genome, resulting in large bubbles, and further resulting in an increase in a large singleton sequence. There are also additional 8 samples showing some large non-ref sequence of a size 2-3 Mb, the situation of fine-scale sequence composition is similar to the HPRC sample but the size is smaller. The rest of CPC samples showed relatively

even-distributed singleton non-ref sequence with a minimum 97.11 Kb, and a median 228.46 Kb.

Figure R 1. the distribution of non-reference singletons.

> Comment 18
About the gnomad variants.

I appreciate that you looked this up and provided the table but I think you missed an opportunity to drive home the main point of your paper. In the summary/abstract you write "The CPC data demonstrate a remarkable increase in discovering novel or missing sequences when individuals are included from underrepresented minority ethnic groups". A devil's advocate might argue that there is less need for a pangenome when short read sequences from a larger group of individuals will capture more small variants and they have already been captured in gnomad. Based on the table you can argue that you add a significant number of variants not previously identified in gnomad and not captured by HPRC, both in the easy and difficult GIAB regions.

Response: Thank the reviewer for the suggestion. We have added a corresponding description in the section "Comparison with HPRC pangenome graph" (lines 873-878, page 12) and a table (S9) in the supplementary information.

Authors' Response to Referee #3

Referee #3 (Remarks to the Author):

In this revision, the authors have addressed most of my concerns. In particular, they constructed a joint pangenome with HPRC and made the pangenome graphs public. The comparison to HPRC makes sense now and is quite interesting. However, the authors haven't addressed all my concerns. I still have one major comment and several minor comments.

Major comment

1) Some of the PAV numbers are likely to be wrong. The authors found 4.72 million SNPs per sample. With short reads, we typically call 3.3-3.5 million SNPs for non-Africans. With long reads we should call more, but 4.72 million is still too large. PAV seems to be aggressively making calls in regions it doesn't have enough power. Probably the majority of additional SNP calls are false positives. At the bottom line, the large difference between PAV-based calls (5.8M) and MC-based calls (4.4M) would confuse readers. I suggest the authors not describe PAV-based small variant calls. If the number of SNP calls is not solid, the number of SV calls with PAV is also questionable. The authors may replace Extended Data Fig 3 with MC-based results. It is ok to describe the SVision results.

Response: We thank the reviewer for this constructive suggestion. We have revised our description about PAV results, to remove the results of variant number and SV length distribution based on PAV calling. The MC-based result of SV size distribution has been shown in the Extended Data Fig 6, thus we have removed the figure of PAV-based SV size distribution, and used SVision results as new Extended Data Fig.

Minor comments

2) I can see the authors have uploaded the contig sequences but they haven't made them public. I trust the authors to release the data once the paper is published.

Response: Thank the reviewer for the trust, indeed, we promise to release the data upon the formal acceptance of our paper.

3) Could the authors run RepeatMasker on unmapped and novel sequences? Probably the majority are centromeric repeats. Would be good to report the number.

Response: We thank the reviewer to point out this. Following the suggestion, we have annotated the unmapped and novel sequences with an integrated pipeline involving *RepeatMasker*, *dna-brnn*, *ETRF* and *SDUST*. In the assembly level, on average 84% of the unmapped sequences are satellite sequences (~39.7-75.1 Mb, 58.1 Mb on average); In the pangenome level, as we have adopted several quality control steps like masking highly repetitive sequences before the construction and clipping out the long unaligned sequences, only 16.5 Mb of the ~190 Mb novel sequences were annotated as the satellite sequences, while most of the other novel sequences (134.8 Mb) were identified only partially repetitive, which is consistent with the results in the HPRC paper. We have added a corresponding description in the section "Assembly of diverse Chinese genomes" (lines 463-465, page 6) and a table in the supplementary information (Table S3).

4) The authors are using inspector to evaluate assemblies, as I suggested earlier. It is worth mentioning that all these tools have false positives, reporting correctly assembled regions as errors. The authors may mention this in the main text. Honestly, I am surprised by the high number. If it were me, I would look at the read alignment in IGV to understand what is happening.

Response: We thank the reviewer to point out this, and we agree with the reviewer that all these tools have false positives. We understand that it is a common challenge and difficulty in the field. We have mentioned this situation in our revised manuscript

as the reviewer suggested (lines 470-473, page 6). In addition, following the suggestion of the reviewer, we have applied IGV to check some regions where assembly errors were reported. We have found an interesting example (Fig. S5B&C) where assessment software cannot determine which assembly is correct when assembly errors are mixed with sequencing errors.

5) The authors found many novel SVs that are close to GWAS hits. It would be good to evaluate the significance of this hits with, for example, permutation test.

Response: We thank the reviewer for the suggestion. We have performed permutation test accordingly, and the updated results can be found in the revised manuscript (lines 983-991, page 15).

Reviewer Reports on the Second Revision:

Referee #2 (Remarks to the Author):

The authors have addressed all my concerns.

Referee #3 (Remarks to the Author):

The authors have addressed my earlier comments. I only have a few discretionary comments, which are all about minor text edits. I am not requesting new materials. I will let the authors decide whether to incorporate them.

1) "We assessed the 116 assemblies with an average total genome length of 3.01 Gb, ranging from 2.88 Gb to 3.12 Gb, and 93.1% of assemblies showed a larger genome length than ungapped GRCh38 (2.94 Gb)". I would not write this way. A female haplotype without chrY is ~3.05 Gb. A male haplotype without chrX is ~2.9 Gb. Because the authors don't have trio, the assembler will have difficulties in phasing. The genome sizes here are not accurate and the sentence may cause confusion.

2) "We found that 16,898 (49.4%) of the novel SVs overlapped the cis-regions (100 kb upstream and downstream of the gene coding regions) of 6,426 protein-coding genes". Typically cis-regulatory elements are upstream to a gene. 100kb also seems large. There are >100kb regulatory regions but in my understanding, most are not this long. Due to the uncertainty in definition, I am not sure if it is a good idea to mention "cis-regions". Number "6,426" is also mentioned in Discussions.

3) "4,344 genes were affected by SVs spanning over 1 kb". It would be good to clarify whether the SVs are disrupting coding regions.

4) Typo: "55.49%%"

Author Rebuttals to Second Revision:

Point-to-point response to Reviewer's comments

Referee #3 (Remarks to the Author):

The authors have addressed my earlier comments . I only have a few discretionary comments, which are all about minor text edits . I am not requesting new materials . I will let the authors decide whether to incorporate them .

1) "We assessed the 116 assemblies with an average total genome length of 3.01 Gb, ranging from 2.88 Gb to 3.12 Gb, and 93.1% of assemblies showed a larger genome length than ungapped GRCh38 (2.94 Gb)" . I would not write this way. A female haplotype without chrY is ~3.05 Gb . A male haplotype without chrX is ~2.9 Gb . Because the authors don't have trio, the assembler will have difficulties in phasing . The genome sizes here are not accurate and the sentence may cause confusion .

Response: We thank the referee for this comment. To address this, we have revised this sentence as "We assessed the 116 assemblies with an average total genome length of 3.01 Gb, ranging from 2.88 Gb to 3.12 Gb due to the size difference between the sex chromosomes, and 93.1% of assemblies showed a larger genome length than ungapped GRCh38 (2.94 Gb)".

2) "We found that 16,898 (49.4%) of the novel SVs overlapped the cis-regions (100 kb upstream and downstream of the gene coding regions) of 6,426 protein-coding genes" . Typically cis-regulatory elements are upstream to a gene . 100kb also seems large . There are >100kb regulatory regions but in my understanding, most are not this long . Due to the uncertainty in definition, I am not sure if it is a good idea to mention "cis-regions" . Number "6,426" is also mentioned in Discussions .

Response: We thank the referee for pointing it out. The term "cis-region" mentioned here does not exclusively refer to cis-regulatory elements. Extensive studies in transcriptional regulation have demonstrated that genetic variations within 100kb (or even 1Mb) regions upstream or downstream of genes can have a major impact on gene expression level. Therefore, this regulatory pattern is defined as cis-regulation, in contrast to trans-regulation, which involves variations located further away from the gene or even on different chromosomes. Following the suggestion, we have modified the terminology here to "nearby-region" to avoid confusion.

3) "4,344 genes were affected by SVs spanning over 1 kb" . It would be good to clarify whether the SVs are disrupting coding regions .

Response: We apologize for the ambiguous description. The 4,344 genes were a subset of the above 6,426 genes which novel SVs overlapped the 100kb nearby regions. We have made corrections and updated the manuscript.

4) Typo: "55 .49%%"

Response: We thank the referee for pointing it out. We have corrected the typo accordingly.